# Structural insights into the activation and inhibition of CXC chemokine receptor 3

Haizhan Jiao[1,4], Bin Pang[2,4], Aijun Liu[1], Qiang Chen[1], Qi Pan[1,3], Xiankun Wang [1], Yunong Xu[1], Ying-Chih Chiang [1]✉, Ruobing Ren [2]✉ & Hongli Hu [1]✉

The chemotaxis of CD4[+] type 1 helper cells and CD8[+] cytotoxic lymphocytes, guided by interferon-inducible CXC chemokine 9–11 (CXCL9–11) and CXC chemokine receptor 3 (CXCR3), plays a critical role in type 1 immunity. Here we determined the structures of human CXCR3–DNG$_i$ complexes activated by chemokine CXCL11, peptidomimetic agonist PS372424 and biaryl-type agonist VUF11222, and the structure of inactive CXCR3 bound to noncompetitive antagonist SCH546738. Structural analysis revealed that PS372424 shares a similar orthosteric binding pocket to the N terminus of CXCL11, while VUF11222 buries deeper and activates the receptor in a distinct manner. We showed an allosteric binding site between TM5 and TM6, accommodating SCH546738 in the inactive CXCR3. SCH546738 may restrain the receptor at an inactive state by preventing the repacking of TM5 and TM6. By revealing the binding patterns and the pharmacological properties of the four modulators, we present the activation mechanisms of CXCR3 and provide insights for future drug development.

The immune cells' migration and positioning are essential for their development and homeostasis, as well as the recruitment and activation in inflammatory response[1,2]. In humans, the movements of various immune cells are coordinated by approximately 50 chemokines and 20 chemokine receptors[1].

The interferon-inducible chemokines CXCL9, CXCL10 and CXCL11 activate chemokine receptor CXCR3 (ref. 3). The expression of CXCR3 is mainly associated with CD4[+] type 1 helper (T$_H$1) cells and CD8[+] cytotoxic lymphocytes, and CXCR3 plays a vital role in T$_H$1 cell-induced inflammatory process[4]. In detail, the CXCR3-mediated peripheralization of CD4[+] T cells in the lymph node is essential for optimal T$_H$1 differentiation[5], and the migration of CD8[+] cytotoxic lymphocyte cells into the infected tissues requires the activation of CXCR3 on the surface of CD8[+] T cells[6–9]. In addition, CXCR3 functions in balancing the generation of effector and memory CD8[+] T cells[10–12], and the rapid peripheralization of memory CD8[+] T cells in lymph nodes on exposure to recall antigens[13–16].

CXCR3 belongs to the class A G-protein-coupled receptor (GPCR) family[1]. CXCL11, CXC10 and CXCL9 bind to CXCR3 with the highest,

intermediate and lowest affinities, respectively[17–19]. Following activation, CXCR3 couples to G$_i$ heterotrimer or β-arrestin to initiate intracellular signals of T cells that express the receptor, causing T cell activation and migration toward the inflamed tissue. The efficacies for G$_i$-mediated signaling, β-arrestin-mediated signaling and receptor internalization of three ligands were systematically probed, and CXCL11 was found to be more biased for the receptor internalization compared with CXCL9 and CXCL10 (ref. 20).

Due to the indispensable role of CXCR3 in T$_H$1-type inflammation, extensive studies of CXCR3 were conducted in infection, autoimmune diseases, transplantation and cancers[9,21–23]. A series of CXCR3 agonists[24–26] and antagonists[27–30] were developed. Peptidomimetic PS372424 is a CXCR3 agonist and was found to inhibit the chemotaxis of activated T cells in the model of rheumatoid arthritis[31]. VUF11222 is one of the first reported nonpeptidomimetic agonists of CXCR3 containing biaryl rings[25].

CXCR3 antagonists were also devoted to inflammatory and autoimmune diseases. For example, SCH546738 has shown significant

[1]Kobilka Institute of Innovative Drug Discovery, School of Medicine, The Chinese University of Hong Kong, Shenzhen, Shenzhen, China. [2]Shanghai Key Laboratory of Metabolic Remodeling and Health, Institute of Metabolism and Integrative Biology, Fudan University, Shanghai, China. [3]School of Life and Health Sciences, School of Medicine, The Chinese University of Hong Kong, Shenzhen, Shenzhen, China. [4]These authors contributed equally: Haizhan Jiao, Bin Pang. ✉e-mail: chiangyc@cuhk.edu.cn; renruobing@fudan.edu.cn; honglihu@cuhk.edu.cn

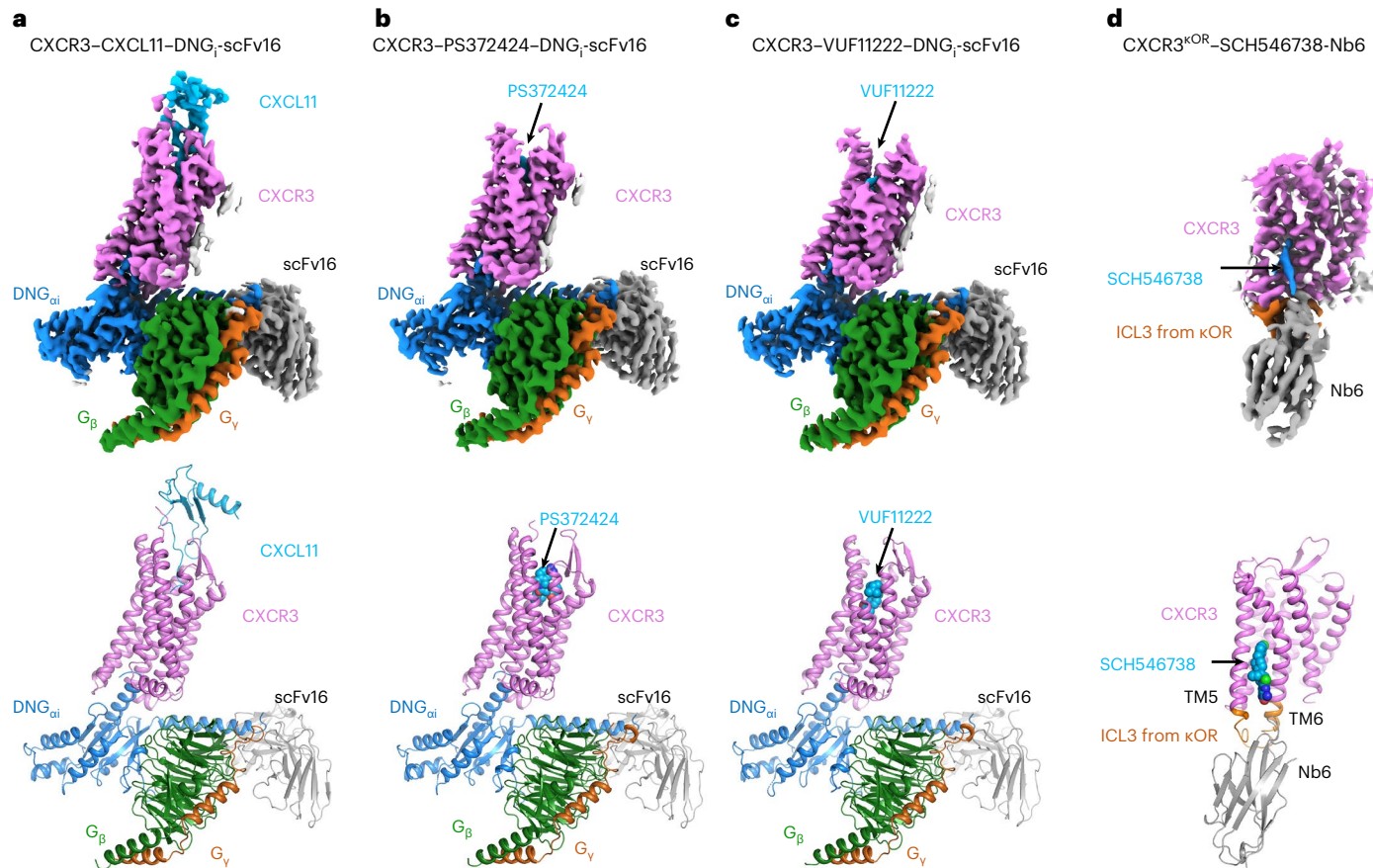

**a** CXCR3–CXCL11–DNGi-scFv16  **b** CXCR3–PS372424–DNGi-scFv16  **c** CXCR3–VUF11222–DNGi-scFv16  **d** CXCR3ᴷᴼᴿ–SCH546738-Nb6

**Fig. 1 | Overall structures of the CXCR3–CXCL11–DNGi-scFv16, CXCR3–PS372424–DNGi-scFv16, CXCR3–VUF11222–DNGi-scFv16 and CXCR3ᴷᴼᴿ–SCH546738-Nb6 complex. a–c**, Density maps and cartoon models of CXCR3–CXCL11–DNGi-scFv16 (**a**), CXCR3–PS372424–DNGi-scFv16 (**b**) and CXCR3–VUF11222–DNGi-scFv16 (**c**). Receptor, ligand, DNGαi, Gβ, Gγ and scFv16 are colored violet, cyan, blue, green, orange and gray, respectively. PS372424 and VUF11222 are shown as spheres. **d**, Density map and cartoon model of CXCR3ᴷᴼᴿ–SCH546738-Nb6. CXCR3, κOR fragment, SCH546738 and Nb6 are colored violet, orange, cyan and gray, respectively. SCH546738 is shown with spheres.

efficacy in several preclinical disease models, to inhibit the activation of T cell chemotaxis with nanomolar potency[30]. However, there is currently no approved drug targeting CXCR3 in the clinic. The molecular basis of CXCR3 ligand recognition and receptor activation is elusive. To address these questions, we determined the structures of CXCR3 activated by three agonists of different types: chemokine CXCL11, peptidomimetic PS372424 and biaryl-type molecule VUF11222. Together with the structure of CXCR3 inhibited by antagonist SCH546378, a systematic illustration of the mechanism of the activation and inhibition of CXCR3 was presented.

## Results

### Overall structures of CXCR3 complexes

The NanoBit tethering strategy was used to stabilize the CXCR3–DNGi complexes, with the LgBit and HiBit fragments fused to the C terminus of CXCR3 and Gβ subunit, respectively[32]. The CXCR3–CXCL11–DNGi complex was obtained by coexpression of CXCR3-LgBit, CXCL11, DNGαi and Gβγ-HiBit, and further stabilized by adding a single chain variable antibody fragment scFv16 after purification by affinity chromatography (Extended Data Fig. 1a–c). For the complex of CXCR3–PS372424–DNGi and CXCR3–VUF11222–DNGi, PS372424 and VUF11222 were supplemented to all the buffers in the purification procedure, respectively (Extended Data Fig. 1d–i). Single particle analysis of datasets collected on cryo-EM finally yields density maps at a nominal global resolution of 3.0, 3.0 and 2.9 Å for complexes CXCR3–CXCL11–DNGi-scFv16, CXCR3–PS372424–DNGi-scFv16 and CXCR3–VUF11222–DNGi-scFv16, respectively (Fig. 1a–c, Extended Data Fig. 2a–c and Table 1). In three density maps, the densities of the receptor, DNGi, scFv16 and the ligands could be well distinguished, and the atomic models were built to a credible level (Fig. 1a–c). For CXCL11, the main chain of the three β sheets and the C-terminal helix could be traced, while for the N terminus and the 30s loop, the densities of side chains could also be well resolved (Extended Data Fig. 3a). For PS372424 and VUF11222, the coordinates generated in AceDRG using SMILE string match well with the density map (Extended Data Fig. 3b,c).

To facilitate the determination of the antagonist bound CXCR3 structure through cryo-EM single particle analysis, a chimeric protein CXCR3ᴷᴼᴿ was generated by replacing the ICL3 of CXCR3 with the ICL3 from κOR[33]. Nanobody Nb6, which recognizes the ICL3 of κOR, was coupled with the chimeric protein CXCR3ᴷᴼᴿ after purification in a buffer containing antagonist SCH546738 (Extended Data Fig. 1j–l). CXCL11 activates CXCR3ᴷᴼᴿ with a potency similar to that of the wild-type CXCR3 (Extended Data Fig. 4a), suggesting that introducing the ICL3 from κOR has little effect on the receptor function. Single particle analysis of the cryo-EM dataset finally yields a density map at a nominal global resolution of 3.6 Å for complex CXCR3ᴷᴼᴿ–SCH546738-Nb6 (Fig. 1d, Extended Data Fig. 2d and Table 1). Residues 54–323 could be traced in the density map, and the extra density between TM5 and TM6 was found to be suitable for accommodating the antagonist SCH546738 (Extended Data Fig. 3d).

### Interactions between CXCR3 and chemokine CXCL11

The proximal 16 amino acid residues of the CXCR3 N terminus were reported to be critical for the recognition of CXCL11 (ref. 34). However,

**Table 1 | Statistics of data collection, data processing, model refinement and validation**

|  | CXCR3–CXCL11–DNG$_i$-scFv16 (EMDB-34914) (PDB 8HNK) | CXCR3–PS372424–DNG$_i$-scFv16 (EMDB-34915) (PDB 8HNL) | CXCR3–VUF11222–DNG$_i$-scFv16 (EMDB-34916) (PDB 8HNM) | CXCR3$^{kOR}$–SCH546738-Nb6 (EMDB-34917) (PDB 8HNN) |
|---|---|---|---|---|
| **Data and processing** | | | | |
| Magnification | 105,000 | 105,000 | 105,000 | 105,000 |
| Voltage (kV) | 300 | 300 | 300 | 300 |
| Electron exposure (e$^-$/Å$^2$) | 56.25 | 56.41 | 54.31 | 53.43 |
| Defocus range (μm) | −1.2 to −1.8 | −1.2 to −1.8 | −1.2 to −1.8 | −1.5 to −2.0 |
| Pixel size (Å) | 0.85 | 0.85 | 0.85 | 0.425 |
| Symmetry imposed | $C1$ | $C1$ | $C1$ | $C1$ |
| Initial particle images (no.) | 7,441,020 | 3,880,618 | 3,501,890 | 15,204,748 |
| Final particle images (no.) | 96,877 | 389,182 | 162,856 | 509,297 |
| Map resolution (Å) | 3.0 | 3.0 | 2.9 | 3.6 |
| FSC threshold | 0.143 | 0.143 | 0.143 | 0.143 |
| Map resolution range | 2.8–6.0 | 2.8–5.0 | 2.8–6.0 | 3.2–5.0 |
| **Refinement** | | | | |
| Initial model used (PDB code) | 6LFO | 6LFO | 6LFO | 6LFL |
| Model resolution (Å) | 3.1 | 3.1 | 3.1 | 3.7 |
| FSC threshold | 0.5 | 0.5 | 0.5 | 0.5 |
| Map sharpening $B$ factor (Å$^2$) | 75.18 | 99.22 | 73.85 | 195.4 |
| Model composition | | | | |
| Nonhydrogen atoms | 9,494 | 8,945 | 8,785 | 3,063 |
| Protein residues | 1,210 | 1,135 | 1,117 | 393 |
| Ligands | 2 | 3 | 3 | 2 |
| $B$ factor (Å$^2$) | | | | |
| Protein | 77.54 | 69.83 | 83.85 | 93.99 |
| Ligand | 90.69 | 107.05 | 119.33 | 66.81 |
| R.m.s. deviations | | | | |
| Bond lengths (Å) | 0.007 | 0.007 | 0.007 | 0.005 |
| Bond angles (°) | 0.660 | 0.667 | 0.673 | 1.026 |
| **Validation** | | | | |
| MolProbity score | 1.74 | 1.67 | 1.80 | 1.82 |
| Clash score | 8.10 | 6.45 | 8.38 | 7.32 |
| Rotamer outliers (%) | 0 | 0 | 0.53 | 0 |
| Ramachandran plot | | | | |
| Favored | 95.72 | 95.53 | 95.00 | 93.78 |
| Allowed | 4.28 | 4.47 | 5.00 | 6.22 |
| Outliers | 0 | 0 | 0 | 0 |

the interactions could not be specified in the structure of CXCR3–CXCL11–DNG$_i$-scFv16 as the densities of the N terminus (residues 1–39) of CXCR3 were not observed. The interactions between the proximal N terminus of the chemokine receptor (known as chemokine recognition site 1.0, CRS1.0) and the core domain of the chemokine (known as chemokine site 1.0, CS1.0) have been widely studied in the chemokine receptor family. In the case of CXCR2, although the interaction between CRS1.0 and CS1.0 has been demonstrated, the N terminus of CXCR2 was not traced in the Cryo-EM structure of CXCR2 complexed with CXCL8 (ref. 35). To investigate the interaction between the proximal N terminus of CXCR3 and CXCL11, coarse-grained molecular dynamics simulations were performed. CXCL11 recruitment could be observed in half of 20 independent

coarse-grained simulations, and one of them is presented in Extended Data Fig. 5. Therefore, the N terminus of CXCR3 may play a key role in the initial recruitment of CXCL11. The conserved Pro42-Cys43 (PC) motif, also termed chemokine recognition site 1.5 (CRS1.5), stretches into the groove formed by β2, β3 and N-loop of CXCL11 (Fig. 2a). The disulfide bond between Cys43$^{N-term}$ and Cys290$^{7.25}$ helps to fix the N terminus of CXCR3 (Fig. 2a).

The N terminus (residues 1–8) of CXCL11, known as the chemokine site 2 (CS2), inserts deeply into the central binding pocket (chemokine recognition site 2, CRS2) of CXCR3 (Fig. 2a). The electric potential of the ligand binding pocket in the receptor is highly negative, making it suitable for binding the positively charged N terminus of chemokine CXCL11 (Fig. 2a). The eight residues (residues 1–8) in the N terminus

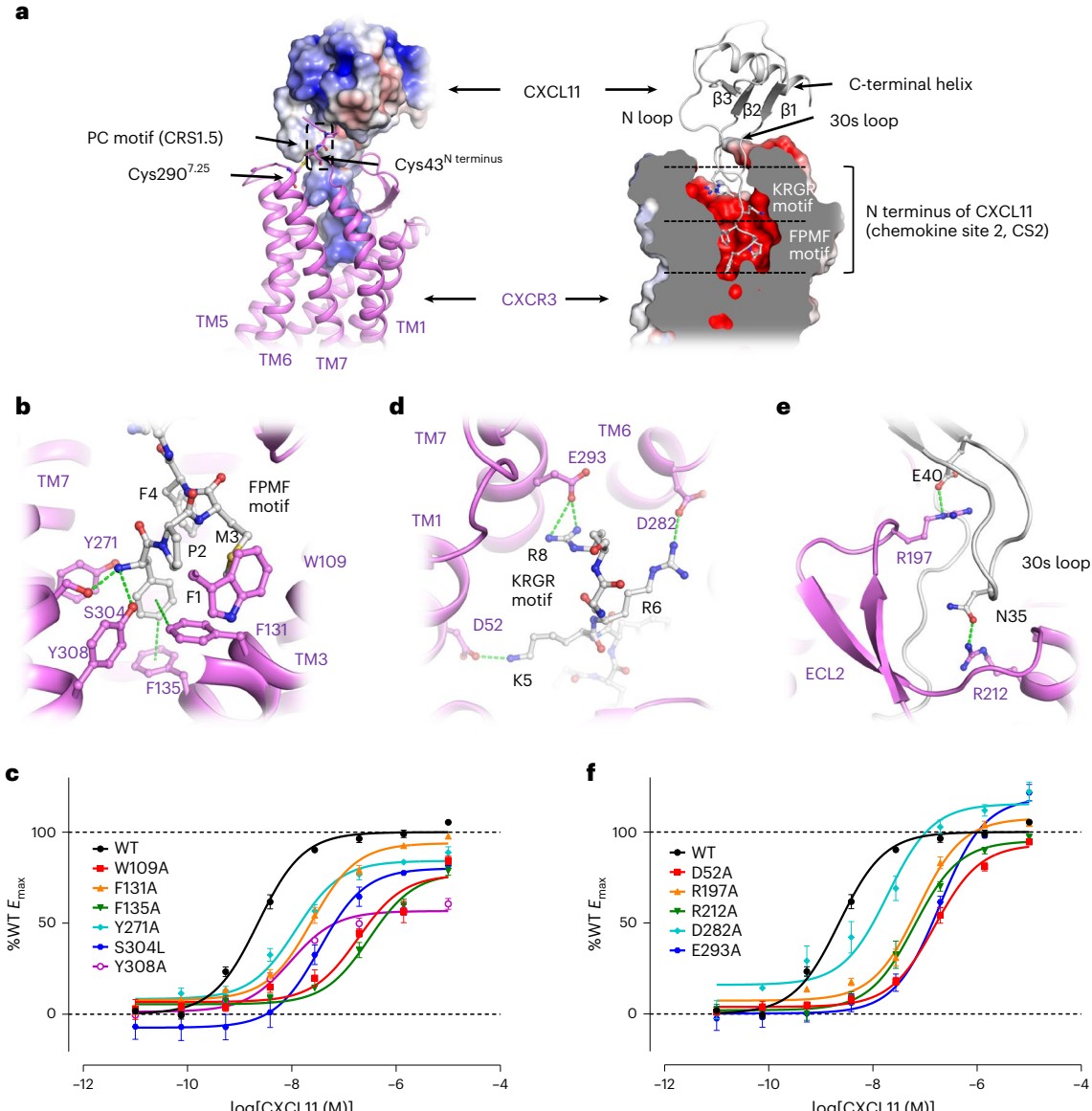

**Fig. 2 | Interactions between CXCR3 and chemokine CXCL11. a**, A general view of the binding pattern of CXCL11. In the left panel, the surface model of CXCL11 (colored by electronic potential) and the cartoon model of CXCR3 (colored violet) are presented. In the right panel, the cartoon model of CXCL11 (colored gray) and the surface model of CXCR3 (colored by electronic potential) are presented. The PC motif and the disulfide bond between Cys43[N-term] and Cys290[7.25] are shown as sticks in the left panel. The FPMF motif and the KRGR motif buried in the central binding pocket are lined out in the right panel. **b**, Interactions between the FPMF motif and CXCR3. **c**, cAMP responses of CXCR3 mutants to CXCL11. **d**, Interactions between the KRGR motif and CXCR3.

**e**, Interactions between the 30s loop in CXCL11 and the ECL2 in CXCR3. In **b**, **d** and **e**, CXCL11 and CXCR3 are colored gray and violet, respectively. The residues involved in interactions are shown as sticks, and the interactions between residues are indicated with green dashes. **f**, cAMP responses of CXCR3 mutants to CXCL11. In **c** and **f**, cAMP responses are normalized to the percentage agonist activity of wild-type CXCR3 (%WT). The data represent means ± s.e.m. ($n = 4$ independent experiments for wild-type CXCR3, $n = 3$ independent experiments for CXCR3 mutants). The expression level of CXCR3 mutants is shown in Extended Data Fig. 4b and the corresponding EC$_{50}$ is summarized in Extended Data Table 1.

of CXCL11 could be divided into an uncharged FPMF motif and a positively charged KRGR motif. The FPMF motif in the proximal N terminus of CXCL11 adopts a torsional posture in the bottom of the pocket, with the side chain of Phe1[CXCL11] inserted most deeply (Fig. 2a, right). The benzene ring of Phe1[CXCL11] forms π–π interactions with Phe131[3.32] and Phe135[3.36] in TM3, and the main chain amino group of Phe1[CXCL11] is hydrogen bonded to the hydroxy group of Ser304[7.39] and Tyr308[7.43] in TM7 (Fig. 2b). Tyr271[6.51] and Trp109[2.60] contribute to stabilizing the FPMF motif through hydrophobic stacking (Fig. 2b). Previous studies indicated that truncation of dipeptide in the FPMF motif results in reduced chemotactic potency, while truncation of three or four

residues in the FPMF motif results in loss of the agonistic activity[36–39]. Therefore, the FPMF motif in CXCL11 plays a key role in receptor activation. Mutation of Trp109[2.60], Phe131[3.32], Phe135[3.36], Tyr271[6.51], Ser304[7.39] and Tyr308[7.43] in CXCR3 results in a reduced CXCL11-induced signaling response (Fig. 2c, Extended Data Fig. 4b and Extended Data Table 1), confirming the essential roles of these residues in CXCL11 binding and receptor activation.

Following the FPMF motif, the KRGR motif of CXCL11 stretches along the upper pocket through extensive charge–charge interactions (Fig. 2a, right), including the salt bridges between Lys5[CXCL11] and Asp52[1.31], Arg6[CXCL11] and Asp282[6.62], Arg8[CXCL11] and Glu293[7.28] (Fig. 2d).

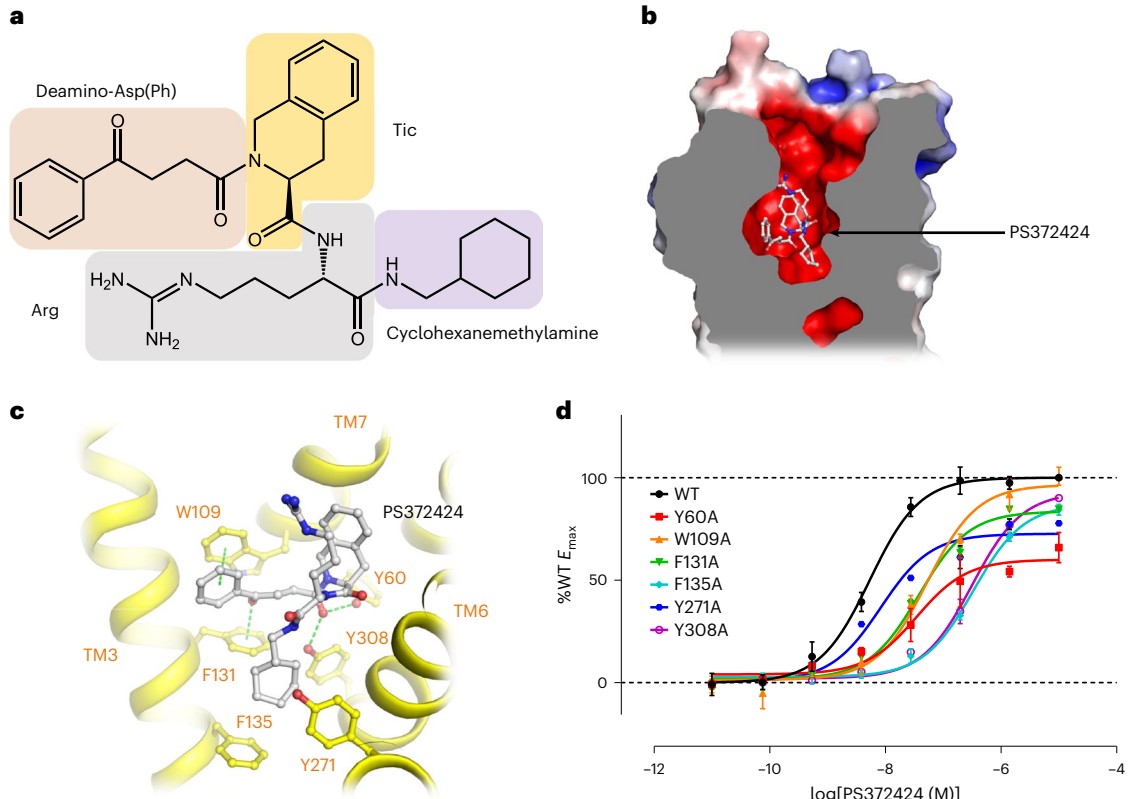

**Fig. 3 | Interactions between CXCR3 and peptidomimetic agonist PS372424.**
**a**, The chemical structure of PS372424. **b**, A general view of the PS372424 binding site in CXCR3. PS372424 and CXCR3 are shown as stick model (colored gray) and surface model (colored by electronic potential), respectively.
**c**, Interactions between PS372424 (gray) and CXCR3 (yellow). PS372424 and the residues involved in interactions are shown as sticks. Green dashes indicate interactions. **d**, cAMP responses of CXCR3 mutants to PS372424. cAMP responses are normalized to the percentage agonist activity of wild-type CXCR3. The data represent means ± s.e.m. ($n$ = 3 independent experiments). The expression level of CXCR3 mutants is shown in Extended Data Fig. 4b and the corresponding $EC_{50}$ is shown in Extended Data Table 1.

The KRGR motif has been proved to be critical for CXCR3 binding, as mutation of KRGR to AAGA significantly reduces the binding affinity[40]. Mutation of Asp52[1.31], Asp282[6.62] and Glu293[7.28] to alanine results in 69-, 8.6- and 85-fold increase in the half-maximum effective concentration ($EC_{50}$) of CXCL11 (Fig. 2f, Extended Data Fig. 4b and Extended Data Table 1), suggesting that the salt bridge interactions between the KRGR motif and the receptor are crucial for CXCL11-induced receptor activation.

Besides CRS1.5 and CRS2, interactions between the 30s loop of CXCL11 and ECL2 of CXCR3 are observed. In detail, the hydrogen bond interaction between Asn35[CXCL11] and Arg212[5.35] positions the 30s loop of CXCL11 on the top of ECL2 (Fig. 2e). Also, Arg197[ECL2] sticks into the bottom of the 30s loop, forming interactions with Glu40[CXCL11] in CXCL11 (Fig. 2e). Mutation of Arg197[ECL2] and Arg212[5.35] results in reduced CXCL11 binding affinity (Fig. 2f, Extended Data Fig. 4b and Extended Data Table 1), suggesting the importance of the interactions between the 30s loop of CXCL11 and ECL2 of CXCR3 in receptor activation.

### Interactions between CXCR3 and agonist PS372424

PS372424 is a peptidomimetic agonist of CXCR3, consisting of a cyclohexanemethylamine group, a natural amino acid Arg, an unnatural amino acid Tic and a deamino-Asp(Ph) group (Fig. 3a)[24]. In the structure of CXCR3–PS372424–DNG$_i$-scFv16, PS372424 is buried deeply in the bottom of the pocket in CXCR3 (Fig. 3b), with the cyclohexane ring surrounded by Phe131[3.32], Phe135[3.36], Tyr271[6.51] and Tyr308[7.43] through hydrophobic stacking (Fig. 3c). The α-carboxy group of deamino-Asp(Ph) is hydrogen bonded to the side chain of Tyr60[1.39] and Tyr308[7.43]. The benzene ring of deamino-Asp(Ph) forms a π–π interaction with the

side chain of Trp109[2.60], and the carbonyl group in the side chain of deamino-Asp(Ph) points against the benzene ring of Phe131[3.32] (Fig. 3c). Mutation of Tyr60[1.39], Trp109[2.60], Phe131[3.32], Phe135[3.36] and Tyr308[7.43] to alanine increased the $EC_{50}$ of PS372424 from 6.5- to 62-fold (Fig. 3d, Extended Data Fig. 4b and Extended Data Table 1), indicating the importance of these residues in PS372424 binding and receptor activation.

In a previous study, three-dimensional (3D) alignments of PS372424 to pentapeptides of CXCL10 suggest that PS372424 may mimic the residues 35–39 or 19–23 of CXCL10 (Extended Data Fig. 6a)[41]. In our structures, the binding site and pose of PS372424 are very similar to the N-terminal pentapeptide of CXCL11. PS372424 occupies the same binding pocket as Phe1[CXCL11], Pro2[CXCL11], Met3[CXCL11] and Lys5[CXCL11] of CXCL11, while Phe4[CXCL11] of CXCL11 occupies an additional minor pocket in the receptor (Extended Data Fig. 6b,c). The cyclohexane group of PS372424 adopts a similar position compared to the side chain of Phe1 of CXCL11 (Extended Data Fig. 6b,c). In conclusion, we suggest that the binding pattern of PS372424 and the N-terminal pentapeptides of CXCL11 share similarities.

### Interactions between CXCR3 and biaryl-type agonist VUF11222

VUF11222 is a small molecular agonist of CXCR3, containing a biaryl group with an ortho-position halogen atom Br on the outer ring and a bicycloaliphatic group (Fig. 4a)[25]. The whole molecule of VUF11222 is encompassed in the bottom half of the pocket (Fig. 4b), with the biaryl group facing downward and the bicycloaliphatic group curling toward TM1. The outer ring of the biaryl group sits on top of Trp268[6.48] and is sandwiched between Phe135[3.36] and Gly307[7.42] (Fig. 4c). The π–π interaction between Phe135[3.36] and the outer ring contributes to

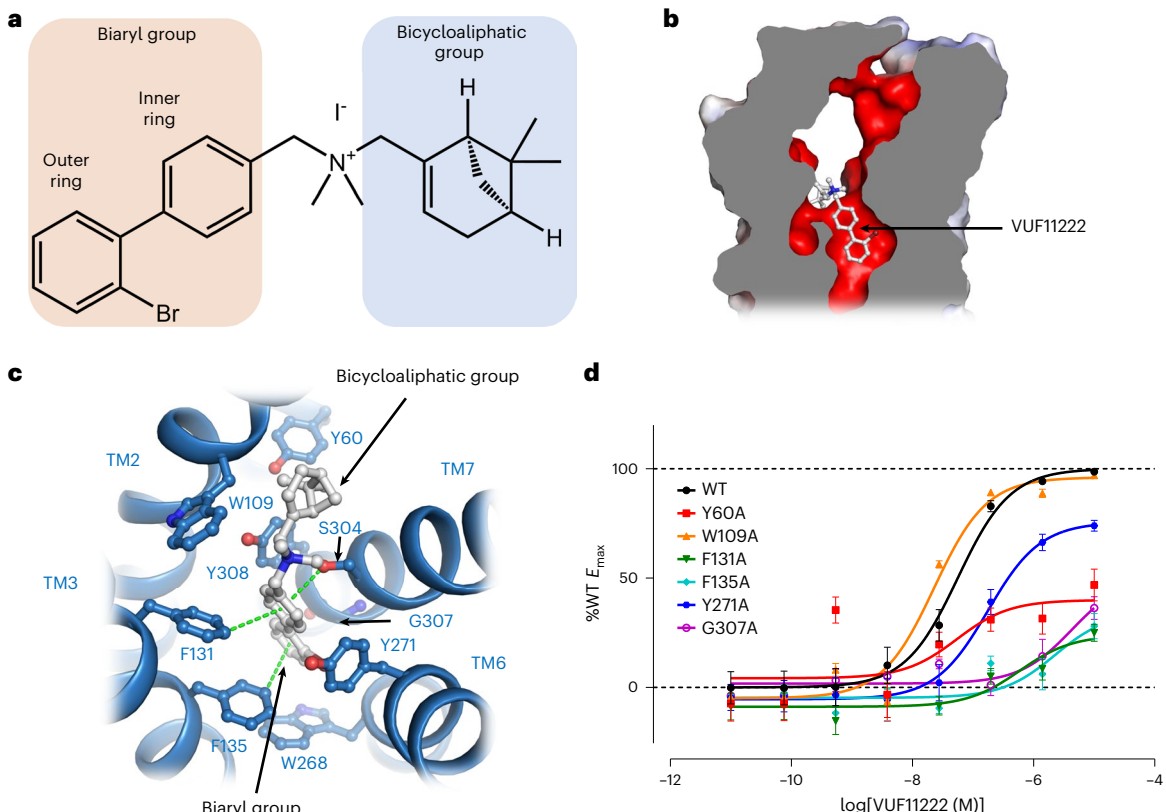

**Fig. 4 | Interactions between CXCR3 and biaryl-type agonist VUF11222. a**, The chemical structure of VUF11222. **b**, A general view of the VUF11222 binding site in CXCR3. VUF11222 and CXCR3 are shown as a stick model (colored gray) and surface model (colored by electronic potential), respectively. **c**, Interactions between VUF11222 (gray) and CXCR3 (blue). VUF11222 and the residues involved in interactions are shown with sticks. Green dashes indicate interactions.

**d**, cAMP responses of CXCR3 mutants to VUF11222. cAMP responses are normalized to the percentage agonist activity of wild-type CXCR3. The data represent means ± s.e.m. ($n = 4$ independent experiments). The expression level of CXCR3 mutants is shown in Extended Data Fig. 4b and the corresponding $EC_{50}$ is shown in Extended Data Table 1.

anchoring the outer ring. The halogen atom Br orientates toward the side chain of Tyr271[6.51]. The inner ring of the biaryl group is surrounded by Tyr271[6.51], Tyr308[7.43], Phe131[3.32] and Ser304[7.39] (Fig. 4c). The benzene ring of Phe131[3.32] forms a π–π interaction, and the hydroxy group of Ser304[7.39] forms an anion–π-like interaction with the inner ring of VUF11222. In contrast to the deeply buried biaryl group, the bicycloaliphatic group only forms weak hydrophobic interactions with the side chains of Trp109[2.60] and Tyr60[1.39] (Fig. 4c). The tight packing between the biaryl group of VUF11222 and CXCR3 suggests that the biaryl group, but not the bicycloaliphatic group, majorly contributes to the activation of the receptor. Mutation of Phe131[3.32], Phe135[3.36], Tyr271[6.51] and Gly307[7.42] to alanine reduced the efficacies of VUF11222-induced signaling response (Fig. 4d, Extended Data Fig. 4b and Extended Data Table 1), suggesting these residues play critical roles in VUF11222 binding and receptor activation. As comparison, mutation of Trp109[2.60] and Tyr60[1.39] results in no obvious changes in the EC50 values (Fig. 4d, Extended Data Fig. 4b and Extended Data Table 1). Notably, the mutation of Tyr60[1.39] significantly reduced the $E_{max}$ of VUF11222, suggesting that the mutation may change the conformational landscape of the receptor or the binding kinetics of the ligand, perhaps by affecting the dissociation rate constant $K_{off}$.

## Activation of CXCR3 by agonists of different types

The binding affinities of CXCL11, PS372424 and VUF11222 are reported to be 0.3, 42 and 63 nM, respectively[17,24,25]. Analysis carried out using PISA (for proteins, interfaces, structures and assemblies)[42] shows that the buried surface areas are 1,473.7 Å[2] for CXCL11, 521.0 Å[2] for PS37242

and 437.2 Å[2] for VUF11222. Therefore, for these three agonists, a larger buried surface area observed in the structure correlates well with a higher binding affinity.

CXCL11 and PS372424 insert into the ligand binding pocket to a similar depth, while VUF11222 inserts about 4.5 Å deeper (Fig. 5a). The binding patterns of CXCL11 and PS372424 are also very similar, with all TMs of CXCR3 having the potential to be aligned well, and only a subtle conformational difference in the ECL2 loop could be observed (Fig. 5b). The TMs of VUF11222-activated CXCR3 go through a more distinct conformational change than CXCL11- or PS372424-activated CXCR3 (Fig. 5c). The root-mean-square deviation (r.m.s.d.) between PS372424- and CXCL11-activated CXCR3 is 0.398 Å, while the r.m.s.d. between VUF11222- and CXCL11/PS372424-activated CXCR3 is 0.938/0.984 Å. According to the density maps, the entire TM1 helix was modeled in the structure of CXCR3–CXCL11–DNGi-scFv16 or the CXCR3–PS372424–DNGi-scFv16 complex, while only the bottom half (residues 60–79) of the TM1 helix was modeled in the structure of CXCR3–VUF11222–DNGi-scFv16 complex (Extended Data Fig. 3a–c). The blurred density of the N terminus of TM1 in the VUFF11222-coupled receptor indicates that the region is highly flexible and is not involved in the binding of VUF11222. Compared with the VUF11222-coupled receptor, a kink in TM1 could be observed in the CXCL11-coupled receptor, making the N terminus of TM1 bend toward TM7 (Fig. 5c). In addition, outward displacements of TM2, TM3 and ECL2 could be observed and may be essential for making room for the insertion of CXCL11 (Fig. 5c).

The side chains of Trp109[2.60], Trp268[6.48] and Tyr308[7.43] are in different conformations in VUF11222-activated CXCR3 compared with

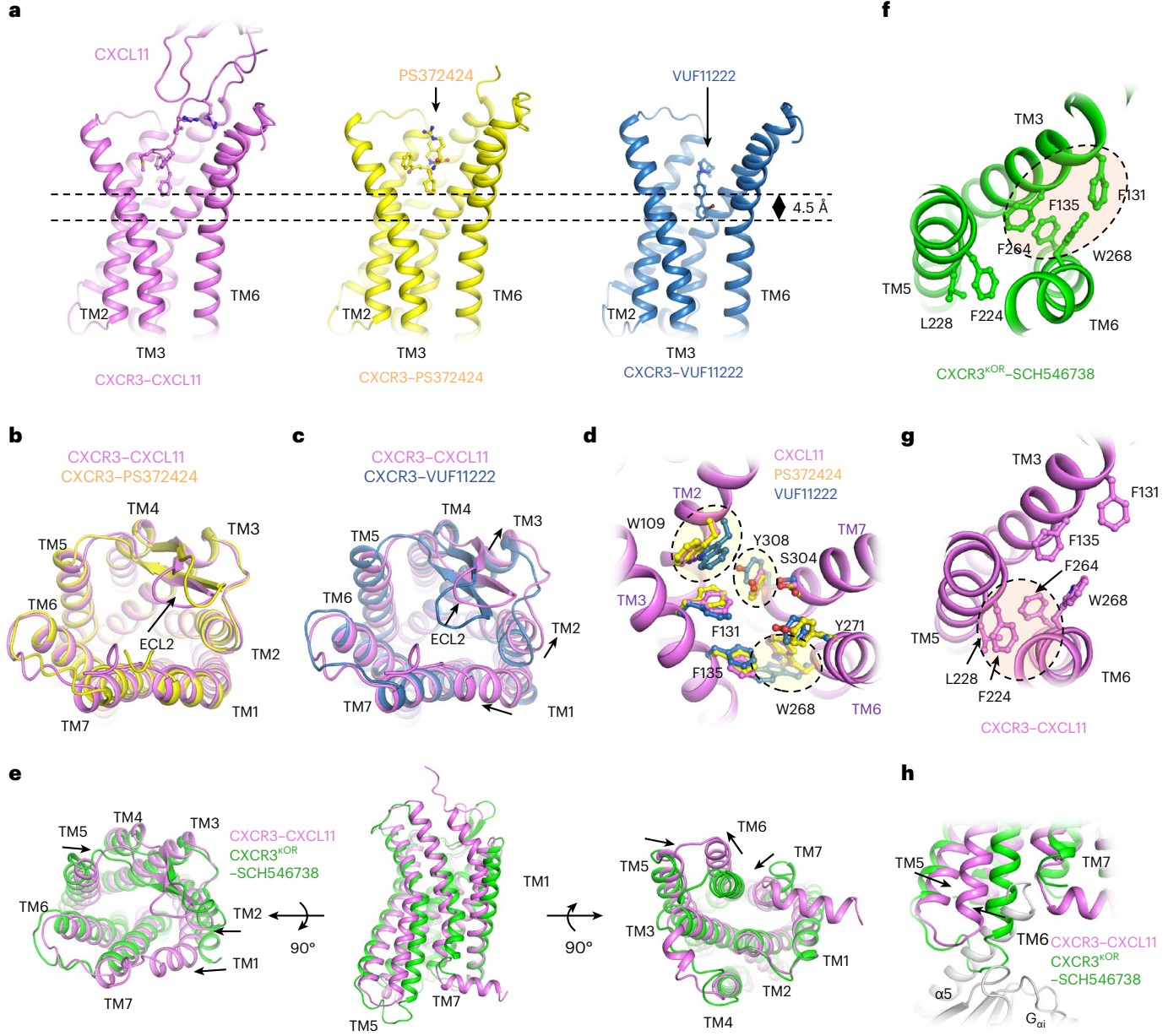

**Fig. 5 | Activation of CXCR3 by agonists of different types. a**, Comparison of the insertion depth of the three agonists. The receptors are shown as cartoons and the ligands are shown as sticks. CXCL11, PS372424 and VUF11222 are colored violet, yellow and blue, respectively. **b**, Superposition of CXCL11-activated CXCR3 (violet) and PS372424-activated CXCR3 (yellow). **c**, Superposition of CXCL11-activated CXCR3 (violet) and VUF11222-activated CXCR3 (blue). **d**, The residues in CXCR3 involved in the interactions with CXCL11 (violet), PS372424 (yellow) and VUF11222 (blue). The residues are shown as sticks, and the CXCL11-activated CXCR3 are shown as cartoons. **e**, Comparison of CXCL11-activated CXCR3 (violet) and SCH546378-inhibited CXCR3 (green). **f**, The packing between TM3 and TM6 in the CXCR3 inhibited by antagonist SCH546738. CXCR3 is shown as a cartoon model and colored green and residues involved in TM packing are shown as sticks. **g**, The packing between TM5 and TM6 in the CXCR3 activated by chemokine CXCL11. CXCR3 is shown as a cartoon model and colored violet, and residues involved in TM packing are shown as sticks. **h**, Outward displacement of TM6 on CXCL11 binding leads to the exposure of the $G_{\alpha i}$ binding pocket. The CXCL11-activated CXCR3, SCH546378-inhibited CXCR3 and $G_{\alpha i}$ are shown as cartoon and colored violet, green and gray, respectively.

CXCL11- or PS372424-activated CXCR3 (Fig. 5d). As the inner ring of the binary group in VUF11222 is at a depth level similar to the side chain of Phe1 in CXCL11 and the cyclohexane group in PS372424, the microswitch of Trp268$^{6.48}$ may be essential for accommodating the outer ring of the binary group in VUF11222. These observations suggest that the receptor activation by VUF11222 adopts a mechanism distinct from that of CXCL11 and PS372424.

When compared with inactive structure of CXCR3, the periplasmic side of CXCL11-activated CXCR3 is more compact. Inward displacements of TM1, TM2 and TM5 in the periplasmic side could be observed after CXCL11 binding (Fig. 5e). In the cytoplasmic side, the binding pocket of $G_{\alpha i}$ was exposed by the outward swing of TM6, inward movement of TM7 and displacement of TM5 toward TM6. (Fig. 5e). Elimination of TM3–TM6 contacts and formation of TM3–TM7 and TM5–TM6 contacts are coupled with conformational changes. In detail, CXCL11 binding disturbs the hydrophobic packing between TM3 and TM6 contributed by Phe131$^{3.32}$, Phe135$^{3.36}$, Phe264$^{6.44}$ and Trp268$^{6.48}$, resulting in the packing of Phe264$^{6.44}$ against Phe224$^{5.74}$ and Leu228$^{5.51}$ (Fig. 5f,g). These reorganizations initialize the outward swing of the cytoplasmic end of TM6 and the exposure of the $G_{\alpha i}$ binding pocket (Fig. 5h).

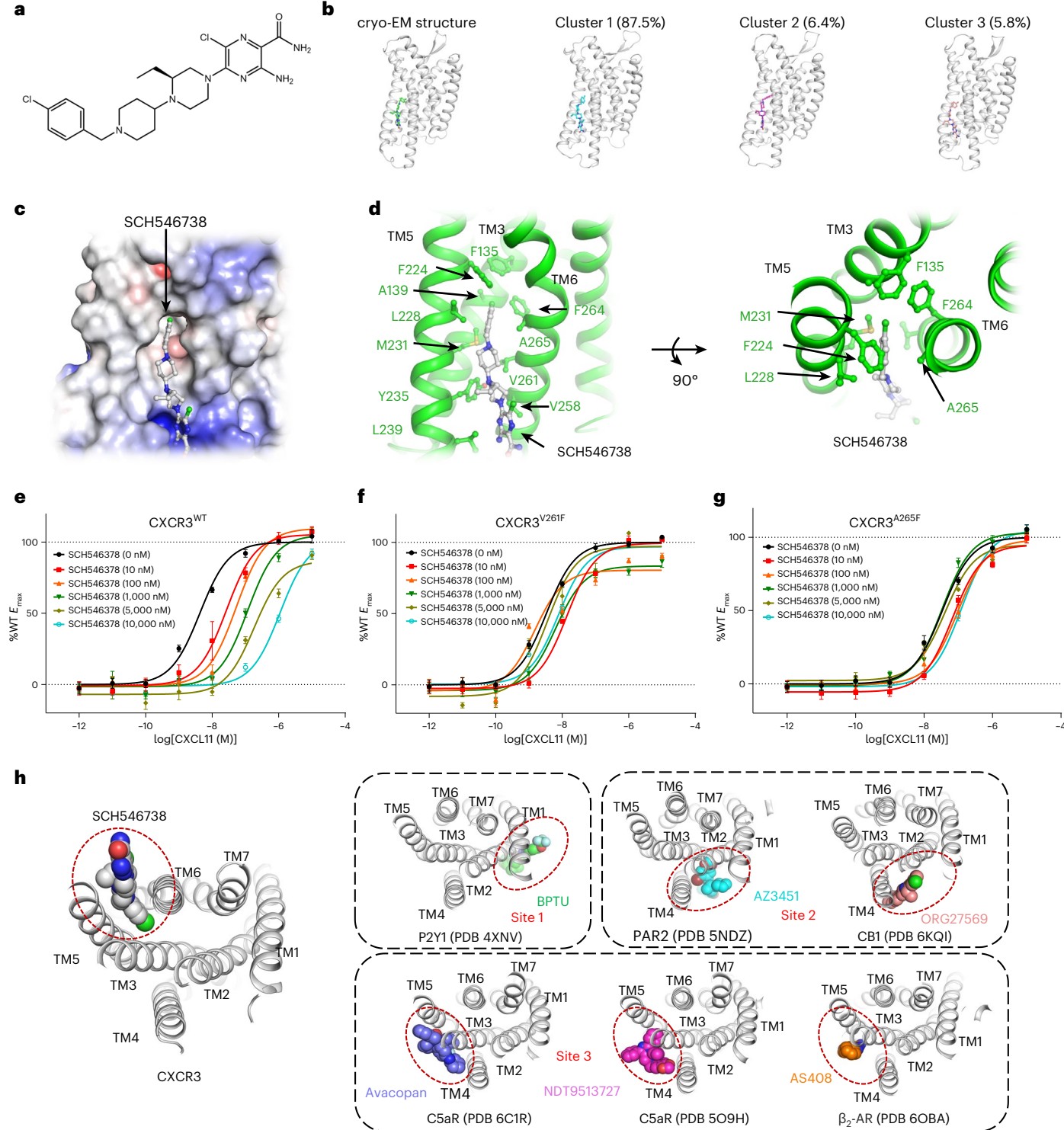

**Fig. 6 | Discovery of an allosteric site in the structure of CXCR3 occupied by SCH546738. a**, The chemical structure of SCH546738. **b**, Molecular dynamics simulation of SCH546738 binding. Shown are the cryo-EM structure and the top three cluster centroids. The size of each cluster is represented as the percentage of the total number of frames in the trajectories. **c**, A general view of the SCH546738 binding pocket in CXCR3. SCH546738 and CXCR3 are shown as a stick model (colored gray) and surface model (colored by electronic potential), respectively. **d**, Interactions between SCH546738 (gray) and CXCR3 (green). SCH546738 and the residues involved in interactions are shown as sticks.

**e**, cAMP responses of CXCR3 to CXCL11 in the presence of SCH546738 at different concentrations. **f**, cAMP responses of CXCR3$^{V261F}$ to CXCL11 in the presence of SCH546738 at different concentrations. **g**, cAMP responses of CXCR3$^{A265F}$ to CXCL11 in the presence of SCH546738 at different concentrations. In **e**–**g**, the data represent means ± s.e.m. ($n = 6$ independent experiments). **h**, Comparison of the allosteric binding sites outside the TMs in class A GPCR. The receptors are shown as cartoons and the antagonists are shown as spheres. Allosteric binding sites are indicated by red circles.

## Discovery of an allosteric site occupied by SCH546738

SCH546738 is a noncompetitive antagonist (Fig. 6a)[30], suggesting that the binding pocket of SCH546738 differs from CXCL11. In the density map of CXCR3[KOR]–SCH546738-Nb6, no continuous density could be observed in the central pocket of CXCR3. By contrast, an omitted density between TM5 and TM6 was found suitable for accommodating the small molecule SCH546738. The refined atomic model of CXCR3[KOR]–SCH546738-Nb6 fits the density map well (Extended Data Fig. 3d). The binding of SCH546738 in the pocket was further evaluated by molecular dynamics simulation. In detail, the CXCR3–SCH546738 complex embedded in a bilayer membrane was simulated with the CHARMM force field[43–45]. The cluster analysis was then applied to analyze the sampled trajectories. The major cluster centroids and the associated cluster sizes are depicted (Fig. 6b). These three clusters represent over 99.7% of the trajectories. While the ligand binds at the allosteric site with poses slightly different from the cryo-EM structure (Fig. 6b), it never drifts away in all molecular dynamics simulations. Hence, the molecular dynamics result supports the binding of SCH546738 in the pocket formed by TM3, TM5 and TM6.

In the structure, SCH546738 is trapped in a hydrophobic pocket surrounded by TM3, TM5 and TM6 (Fig. 6c). The central of the hydrophobic pocket, formed by Phe135[3.36], Ala139[3.40], Phe224[5.47], Leu228[5.51], Met231[5.54], Val261[6.41], Phe264[6.44] and Ala265[6.45], contributes to stabilize the head of SCH546738. The tail of SCH546738 stretches out the central pocket and contacts with Tyr235[5.58], Leu239[5.62] and Val258[6.38] (Fig. 6d). Mutants CXCR3[V261F] and CXCR3[A265F] were constructed to validate the allosteric binding site. Both CXCR3[V261F] and CXCR3[A265F] retain receptor activity. The $EC_{50}$ for CXCR3[V261F] mutant is comparable to that of CXCR3[WT] (Fig. 6e,f and Extended Data Table 2). The $EC_{50}$ for CXCR3[A265F] is about one order higher (Fig. 6e,g and Extended Data Table 2). For the wild-type receptor, increasing in the SCH546738 concentration resulted in a decrease in the potency of CXCL11 (Fig. 6e and Extended Data Table 2). By contrast, for CXCR3[V261F] and CXCR3[A265F], the antagonism of SCH546738 is less evident (Fig. 6f,g and Extended Data Table 2). The assay indicates that mutation of Val261 and Ala265 to phenylalanine may reduce the space of the allosteric binding site and interfere with the binding of SCH546738. As the repacking of TM5-TM6 is critical for receptor activation, the insertion of SCH546738 between TM5 and TM6 may prevent the process and constraint the receptor at an inactive state.

## Discussion

The allosteric sites close to the TMs of the class A GPCR could be divided into three classes: BPTU binds to P2Y1 (Protein Data Bank (PDB) code 4XNV) in site 1 that was formed mainly by TM1–TM2–TM3, AZ3451/ORG27569 (PDB 5NDZ/6KQI) binds to PAR2/CB1 in site 2 that was formed mainly by TM2–TM3–TM4 and avacopan/NDT9513727/AS408 binds to C5aR/C5aR/β₂-AR (PDB 6C1R/5O9H/6OBA) in site 3 that was formed mainly by TM3–TM4–TM5 (Fig. 6h)[46–51]. The allosteric binding pocket between TM3, TM5 and TM6 in CXCR3 is distinct from these known allosteric sites in the spatial position (Fig. 6h). The discovery of an allosteric site in CXCR3 different from the other allosteric sites in class A GPCR may assist in developing new allosteric antagonists targeting CXCR3. A structure with higher resolution is still necessary for revealing the precise interactions between CXCR3 and SCH546738.

By superposition of CXCR3–CXCL11 with CXCR2–CXCL8 (PDB 6LFO), CCR5–CCL5 (PDB 7F1R), CCR5–CCL3 (PDB 7F1Q), CCR6–CCL20 (PDB 6WWZ) and ACKR3–CXCL12 (PDB 7SK5)[35,52–54], we found that the N terminus of CXCL11 is in the deepest position, while the N terminus of CCL20 is in the shallowest position (Extended Data Fig. 7a). The maximum depth difference of these six chemokines is about 11.4 Å. In addition, CXCL11 shares a binding pose similar to CXCL8 and CCL20 with minor deviations. Compared to CXCL11, CCL5 and CCL3 undergo a rotation along a pseudo axis parallel to the membrane surface, while CXCL12 undergoes a rotation along a pseudo axis perpendicular to the membrane surface. The difference in the insertion depth and binding pose suggests that the binding patterns of chemokines differ widely.

CXCR1 and CXCR2 could be activated by ELR⁺ chemokines CXCL1–3 and 5–8, which contain an ELR motif in the N terminus (Extended Data Fig. 7b). When compared with the structure of CXCL8 coupled CXCR2, CXCL11 inserts about 6.3 Å deeper than CXCL8 (Extended Data Fig. 7c). An 8.1° tilt in the chemokine and a 4.2° tilt in the $G_i$ protein could be observed in the structure of CXCR3–CXCL11-$G_i$ when compared with the structure of CXCR2–CXCL8-$G_i$ (Extended Data Fig. 7c) (using the mass centers of the chemokine, the receptor and the $G_i$ complex as reference points). The residues corresponding to the ELR motif in CXCL8 are residues RGR in CXCL11. The main difference is that the negatively charged residue Glu4[CXCL8] is replaced with a positively charged residue Arg6[CXCL11] (Extended Data Fig. 7b). Four charged residues Arg208[5.35], Arg212[5.39], Asp274[6.58] and Arg278[6.62] in CXCR2 cooperate to interact with the residue Glu4[CXCL8] in CXCL8 (Extended Data Fig. 7d,e). The corresponding site of 6.62 is Asp282[6.62] in CXCR3, forming a salt bridge interaction with Arg6[CXCL11] in CXCL11 (Extended Data Fig. 7d,e). Therefore, the difference in charge of residue 6.62 may play a vital role in determining the chemokine selectivity of CXCR2 and CXCR3.

At present, only one structure of chemokine receptor activated by a small molecular agonist was available, which is the structure of atypical chemokine receptor ACKR3 complexed with the small molecule agonist CCX662 (ref. 54). The structures of CXCR3 coupled with small molecule agonists PS372424 and VUF11222 would provide more information for comparing the binding pattern of small molecule agonists with chemokines targeting typical chemokine receptors. These two small molecule agonists activated the receptor differently. The binding mode of PS372424 is similar to the intrinsic agonist CXCL11, while VUF11222 inserts deeper and triggers a distinct conformational change in Trp268[6.48].

According to previous studies, CXCL11 is more efficacious at activating $G_i$ signaling and β-arrestin signaling than CXCL10 and CXCL9, and is slightly biased toward β-arrestin signaling[20,55]. PS372424 is efficacious at inducing receptor internalization, while its analog VUF10661 was reported to recruit β-arrestin with greater efficacy than CXCL11 (ref. 31,56). The biased signaling of VUF11222 has not been studied, but its analog VUF11418 is reported to be a G-protein biased agonist[57,58]. In summary, CXCL11 and PS372424 are potentially β-arrestin biased agonists, while VUF11222 is potentially a G-protein biased agonist.

In the absence of a structure of β-arrestin-coupled CXCR3, it is difficult to interpret the molecular basis of bias signaling of CXCR3. Structural analysis of available structures of β-arrestin-coupled GPCRs may give some clues. When superposed with the structure of β-arrestin-coupled 5HT2B, a micro-tilt in the lower half of TM7 (named TM7[7.47-7.53]) is observed in the structure of $G_q$-coupled 5HT2B (Extended Data Fig. 8a)[59]. Superposition of β-arrestin-coupled with $G_o$-coupled ACM2 and superposition of β-arrestin-coupled with $G_q$-coupled NTR1 show similar micro-tilt in TM7[7.47-7.53] (Extended Data Fig. 8b,c)[60–63]. Therefore, it appears that TM7[7.47-7.53] tilt to a greater degree in the G-protein-bound receptor than in the β-arrestin-bound receptor. The micro-tilt in TM7[7.47-7.53] extends the side chain of 7.53 (a conserved Tyr in the NPxxY motif) into a different spatial position in the intracellular binding cavity. Since residue 7.53 is sandwiched between TM3 and TM6, the spatial position of 7.53 may affect the overall property of the signaling protein binding cavity and ultimately influence the signaling bias.

In our structures, no obvious difference could be observed in the spatial position of 7.53. This is mainly due to the coupling of G protein in all three structures (Extended Data Fig. 8d). However, a micro-tilt at 7.47 is observed in the VUF11222-coupled CXCR3 when superposed with the CXCL11- and PS372424-coupled CXCR3 (Extended Data Fig. 8d). As shown in Extended Data Fig. 8e, the insertion of VUF11222 causes a microswitch in Trp268 that repels His310 away. Together with the repulsion of Ser307 and Tyr308 directly caused by VUF11222, the N terminus of TM7[7.47-7.53] is tilted (Extended Data Fig. 8e). A possible mechanism for

the biased signaling of CXCR3 is shown in Extended Data Fig. 8f. The insertion of VUF11222 initiates the tilt in the N terminus of TM7[7.47-7.53], which is transferred to the C terminus of TM7[7.47-7.53], moves Tyr318[7.53] toward the central of the signaling protein binding cavity and ultimately influences the binding of signaling proteins. A structure of CXCR3 coupled with β-arrestin and extensive assays are required in the future to validate the hypothesis.

## Online content

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

## Methods

### Design and expression of constructs

Wild-type human CXCR3 (residues 1–361) was cloned into pFastBac1 vector with an N-terminal Flag tag after a HA signal peptide. The protein BRIL was fused to the N terminus of CXCR3 to enhance the expression level of CXCR3, and the protein LgBit was fused to the C terminus of CXCR3 to stabilize the CXCR3–DNG$_i$ complex. Domain negative G$_{\alpha i1}$ (DNG$_{\alpha i1}$) was cloned into a pFastBac1 vector and G$_{\beta 1 \gamma 2}$ with HiBit fusion to the C terminus of G$_{\beta 1}$ was cloned into a pFastBac Dual vector. Human CXCL11 with a C-terminal 8×His tag, scFv16 with an N-terminal GP67 signal peptide and a C-terminal 8×His tag were cloned into a pFastBac1 vector, respectively.

To facilitate structure determination of the inactive CXCR3 through the coupling of nanobody Nb6, the ICL3 loop of CXCR3 (residues 241–254) was replaced with the ICL3 loop from the kappa-opioid receptor (residues 252–273). The resulting chimeric protein CXCR3$^{KOR}$ with BRIL fused to the N terminus was cloned into a pFastBac1 vector, and an N-terminal Flag tag after a HA signal peptide was added. Nb6 was cloned into a pFastBac1 vector with an N-terminal GP67 signal peptide for secretion and a C-terminal 8×His tag for purification. The sequence of synthesized CXCR3 and CXCL11 genes are listed in Supplementary Table 1. Primers used for the generation of CXCR3 and its mutant used in cAMP assay are listed in Supplementary Table 2.

Proteins were expressed in Sf9 insect cells (Thermo Fisher Scientific, catalog no. 11496015) using the Bac-to-Bac baculovirus system. For the CXCR3–CXCL11–DNG$_i$ complex, Bril-CXCR3-LgBit, DNG$_{\alpha i1}$, G$_{\beta 1 \gamma 2}$-HiBit and CXCL11 (virus ratio of 1:1:1:2) were coexpressed in Sf9 insect cells at a density of $3.0 \times 10^6$ cells per ml. For the CXCR3–PS372424–DNG$_i$ and CXCR3–VUF11222–DNG$_i$ complex, Bril-CXCR3-LgBit, DNG$_{\alpha i1}$ and G$_{\beta 1 \gamma 2}$-HiBit were coexpressed at a virus ratio 1:1:1. Cells were collected after incubation for 60 h at 27 °C. scFv16 and Nb6 were expressed in Sf9 insect cells at 27 °C for 72 h.

### Purification of CXCR3–CXCL11–DNG$_i$-scFv16, CXCR3–PS372424–DNG$_i$-scFv16 and CXCR3–VUF11222–DNG$_i$-scFv16 complexes

The cell pellets of CXCR3–CXCL11–DNG$_i$ were thawed in 20 mM HEPES-Na, pH 7.5, 300 mM NaCl, 1% LMNG, 0.2% CHS, 10% glycerol, 5 mM MgCl$_2$, 25 mU ml$^{-1}$ apyrase and complete protease inhibitor cocktail. The suspension was homogenized and solubilized for 2 h at 4 °C, followed by centrifugation at 40,000$g$ for 30 min. Then the supernatant was collected and incubated with anti-DYKDDDDK G1 affinity resin (Genscript) for 1.5 h at 4 °C. The resin was washed in buffer 20 mM HEPES-Na, pH 7.5, 100 mM NaCl, 0.1% LMNG, 0.02% CHS, 10% glycerol, 5 mM MgCl$_2$, followed by buffer 20 mM HEPES-Na, pH 7.5, 100 mM NaCl, 0.01% LMNG, 0.002% CHS, 10% glycerol and 5 mM MgCl$_2$. Then, the protein was eluted in buffer 20 mM HEPES-Na, pH 7.5, 100 mM NaCl, 0.01% LMNG, 0.002% CHS, 10% glycerol, 5 mM MgCl$_2$ and 0.25 mg ml$^{-1}$ DYKDDDDK peptide. The protein was concentrated using a 100 kDa molecular mass cutoff concentrator and incubated with scFv16 for 30 min on ice with a molar ratio of 1:1.5. The complex was further purified by size-exclusion chromatography on Superose 6 10/300 Increase column in a running buffer 20 mM HEPES-Na, pH 7.5, 100 mM NaCl, 0.005% LMNG, 0.001% CHS and 5 mM MgCl$_2$. The peak fraction of the complex was collected and concentrated to around 5.0 mg ml$^{-1}$ for cryo-EM sample preparation.

The purification of CXCR3–PS372424–DNG$_i$-scFv16 and CXCR3–VUF11222–DNG$_i$-scFv16 complex was similar to that of the CXCR3–CXCL11–DNG$_i$-scFv16 complex. The only difference is that 4.2 μM PS372424 (MedChemExpress) or 0.72 μM VUF11222 (Tocris Bioscience) was added to all the buffers in protein purification.

### Purification of CXCR3$^{KOR}$–SCH546738-Nb6 complex

The cell pellets of CXCR3$^{KOR}$ were thawed in 20 mM HEPES-Na, pH 7.5, 300 mM NaCl, 1% LMNG, 0.2% CHS, 10% glycerol, 0.1 μM SCH546738 (MedChemExpress) and complete protease inhibitor cocktail. After homogenization and solubilization for 2 h at 4 °C, the suspension was centrifuged at 40,000$g$ for 30 min. The collected supernatant was incubated with anti-DYKDDDDK G1 affinity resin (Genscript) for 1.5 h at 4 °C. Then the resin was washed in buffer 20 mM HEPES-Na, pH 7.5, 100 mM NaCl, 0.1% LMNG, 0.02% CHS, 10% glycerol, 0.1 μM SCH546738, followed by buffer 20 mM HEPES-Na, pH 7.5, 100 mM NaCl, 0.01% LMNG, 0.002% CHS, 10% glycerol and 0.1 μM SCH546738. The protein was eluted in buffer 20 mM HEPES-Na, pH 7.5, 100 mM NaCl, 0.01% LMNG, 0.002% CHS, 10% glycerol, 0.1 μM SCH546738 and 0.25 mg ml$^{-1}$ DYKDDDDK peptide. The protein was concentrated and incubated with Nb6 for 30 min on ice with a molar ratio of 1:1.5, and the CXCR3$^{KOR}$–SCH546738-Nb6 complex was further purified by size-exclusion chromatography on Superose 6 10/300 Increase column in a running buffer 20 mM HEPES-Na, pH 7.5, 100 mM NaCl, 0.005% LMNG, 0.001% CHS and 0.1 μM SCH546738. The peak fraction of the complex was collected and concentrated to around 5.0 mg ml$^{-1}$ for cryo-EM sample preparation.

### Cryo-EM sample preparation and data acquisition

The purified CXCR3–CXCL11–DNG$_i$-scFv16, CXCR3–PS372424–DNG$_i$-scFv16, CXCR3–VUF11222–DNG$_i$-scFv16 or CXCR3$^{KOR}$–SCH546738-Nb6 were applied to glow-discharged 300-mesh alloy grids (CryoMatrix M024-Au300-R12/13, Zhenjiang Lehua Technology) and subsequently vitrified using Vitrobot Mark IV. The images were collected in the counted-Nanoprobe mode on a 300 kV Titan Krios Gi3 electron microscope (Thermo Fisher Scientific) equipped with a Gatan K3 Summit detector and GIF Quantum energy filter.

For CXCR3–CXCL11–DNG$_i$-scFv16, CXCR3–PS372424–DNG$_i$-scFv16 and CXCR3–VUF11222–DNG$_i$-scFv16, all video stacks with 50 frames were collected using SerialEM[64] software at a nominal magnification of ×105,000, a pixel size of 0.85 Å and a defocus range of −1.2 to −1.8 μm.

For CXCR3–CXCL11–DNG$_i$-scFv16, each video stack was recorded for 4.0 s with 0.08 s exposure per frame at a total dose of 56.25 e$^{-}$/Å$^2$. For CXCR3–PS372424–DNG$_i$-scFv16, each video stack was recorded for 2.0 s with 0.04 s exposure per frame at a total dose of 53.79 e$^{-}$/Å$^2$. For CXCR3–VUF11222–DNG$_i$-scFv16, each video stack was recorded for 2.0 s with 0.04 s exposure per frame at a total dose of 54.31 e$^{-}$/Å$^2$.

For CXCR3$^{KOR}$–SCH546738-Nb6, video stacks with 50 frames were collected using SerialEM software at super-resolution mode, with a pixel size of 0.425 Å and a defocus range of −1.5 to −2.0 μm. Each video stack was recorded for 2.0 s with 0.04 s exposure per frame at a total dose of 53.42 e$^{-}$/Å$^2$.

### Cryo-EM data processing of CXCR3–CXCL11–DNG$_i$-scFv16, CXCR3–PS372424–DNG$_i$-scFv16 and CXCR3–VUF11222–DNG$_i$-scFv16

In total, 5,500, 3,185 and 2,891 videos were collected for CXCR3–CXCL11–DNG$_i$-scFv16, CXCR3–PS372424–DNG$_i$-scFv16 and CXCR3–VUF11222–DNG$_i$-scFv16, respectively. Data were processed in RELION v.3.1 (ref. [65]) and CryoSPARC[66]. After correction of the beam-induced motion by MotionCor2 (ref. [67]), contrast transfer function (CTF) parameters were estimated by Gctf[68]. Next, 7,441,020, 3,880,618 and 3,501,890 particles were auto-picked and extracted in a pixel size of 1.70 Å for each dataset. For CXCR3–CXCL11–DNG$_i$-scFv16 and CXCR3–VUF11222–DNG$_i$-scFv16, bad particles were removed through several rounds of 3D classification using the structure of CXCL8-activated CXCR2 (Electron Microscopy Data Bank (EMDB) [0879]) as a reference. While for CXCR3–PS372424–DNG$_i$-scFv16, bad particles were removed through several rounds of two-dimensional (2D) classification in CryoSPARC. The selected particles were re-extracted in the original pixel size of 0.85 Å, and extensive 3D classification without alignment was performed with a mask on the complex excluding scFv16 to distinguish different conformations. Finally, 96,877 particles of CXCR3–CXCL11–DNG$_i$-scFv16 were applied to 3D refinement after CTF refinement and particle polishing, yielding a density map of 3.0 Å resolution; then, 389,182 particles of CXCR3–PS372424–DNG$_i$-scFv16 were applied to 3D refinement after CTF refinement and particle polishing, yielding a

density map of 3.0 Å resolution. For CXCR3–VUF11222–DNG$_i$-scFv16, a density map of 2.9 Å was reconstructed using 162,856 particles after CTF refinement and particle polishing.

### Cryo-EM data processing of CXCR3$^{κOR}$–SCH546738-Nb6
Here, 12,944 videos were collected for CXCR3$^{κOR}$–SCH546738-Nb6. Correction of the beam-induced motion was accomplished by Motion-Cor2 and CTF parameters were estimated by Gctf. In total, 15,204,748 particles were auto-picked using the Laplacian-of-Gaussian method and extracted in a pixel size of 1.70 Å in RELION. Next, the particles were imported into CryoSPARC and 6,800,282 particles were screened out through two rounds of 2D classification. Four initial models were generated in CryoSPARC, and after excessive hetero-refinement in CryoSPARC, 509,297 particles were retained and re-extracted in a pixel size of 0.85 Å in RELION. After CTF refinement and particle polishing, one round of local refinement was performed in CryoSPARC, yielding a density map of 3.6 Å resolution.

### Model building and refinement
Map sharpening was accomplished by Autosharpen in Phenix[69]. The model of CXCR3 predicted by AlphaFold, the model of CXCL11 (PDB 1RJT), the model of G$_i$ ternary complex and scFv16 (from PDB 6LFO) were docked into the density map of CXCR3–CXCL11–DNG$_i$-scFv16 in ChimeraX. Mutations of G203A and A326S were generated in COOT[70] to get DNG$_i$. The model was refined by iterative manual adjustment and rebuilding in COOT and real space refinement in Phenix[71]. The refined model was used as a reference to rescale the amplitude of the map by LocScale[72] in the CCP-EM package, resulting in better connectivity of the density of CXCL11. In the final model of CXCR3–CXCL11–DNG$_i$-scFv16, residues 22–94 of CXCL11 and residues 40–336 of CXCR3 were traced.

For CXCR3–PS372424–DNG$_i$-scFv16, the coordinate of PS372424 was generated by AceDRG[73] using the SMILE string and was docked into the density map in COOT. The structure of the complex was then manually adjusted in COOT and refined in Phenix. In the refined model, residues 42–336 of CXCR3 could be traced.

For CXCR3–VUF11222–DNG$_i$-scFv16, the coordinate of VUF11222 was generated by AceDRG using the SMILE string and was docked into the density map in COOT. The structure of the complex was then manually adjusted in COOT and refined in Phenix. In the refined model, residues 60–336 of CXCR3 could be traced.

For CXCR3$^{κOR}$–SCH546738-Nb6, CXCR3 and Nb6 were docked into the density map in ChimeraX. The ICL3 loop of CXCR3 (residues 241–254) was replaced with the ICL3 loop from the kappa-opioid receptor (residues 252–273) in COOT to get chimeric CXCR3$^{κOR}$. The coordinate of SCH546738 generated by AceDRG was fitted into the density, and then the structure of the complex went through iterative manual adjustment in COOT and real space refinement in Phenix.

The geometries of models were validated using MolProbity[74]. The figures of the structures were prepared in ChimeraX[75] and PyMOL (Schrödinger, LLC).

### Split-luciferase-based cAMP reporter assays
HEK293T cells (ATCC, no. CRL-11268) coexpressing CXCR3 and different mutants (in vector pcDNA3.1) along with a split-luciferase-based cAMP biosensor (GloSensor, Promega) were seeded in 96-well white clear bottom cell culture plates (Beyotime; 15,000 cells per well, 100 µl per well) in DMEM (Macgene) containing 10% fetal bovine serum (Every Green). The next day, the culture medium was removed and 40 µl per well of drug buffer (1×HBSS, 20 mM HEPES, pH 7.4) was added for 2 min at room temperature, followed by the addition of 20 µl of agonist drug solutions (the initial agonist drug concentration is 10 µM, on the basis of 1/7 gradient dilution) for 10 min at room temperature. To measure agonist activity for G$_{αi}$-coupled receptors, 20 µl of drug buffer supplemented with luciferin (25 mg ml$^{-1}$ final concentration) and forskolin (26 µM final concentration) was added. Luminescence intensity was quantified 5 min later.

The expression levels of CXCR3 mutants were determined by flow cytometry as described below. Approximately $1 \times 10^6$ transfected HEK293T cells were collected and washed twice with PBS. Cells were blocked with 5% bovine serum albumin (BSA) at room temperature for 15 min and then incubated with anti-FLAG mouse monoclonal antibody (1:100, Cwbio) in PBS containing 1% BSA at 4 °C for 1 h. Cells were washed three times with PBS and then incubated with antimouse Alexa Fluor 488-labeled goat antimouse IgG (1:300, Beyotime) in PBS containing 1% BSA at 4 °C in the dark for 1 h. After washing twice with PBS, the cells were resuspended in 200 µl of PBS for detection in a BD LSR II flow cytometer. Approximately 10,000 cellular events were counted for each sample, and the fluorescence intensity data were collected.

### Coarse-grained simulations
The missing residues 1–39 of CXCR3 were modeled using AlphaFold structure, while the CXCL11 was placed at least 30 Å away from CXCR3 structure. The system was mapped into the ElNeDyn22 coarse-grained model using the CHARMM-GUI Martini Maker[76]. The system was embedded in a 150 × 150 Å membrane, which was composed of 90% 1,2-dioleoyl-sn-glycero-3-phosphocholine and 10% cholesterol lipids. The system was then further solvated in salt water of 150 mM salt concentration. Before the production run, a 5,000-step energy minimization and a restrained 5 ns NPT (number of particles, pressure, temperature) equilibration were performed. Then, a 1 µs production run was performed using the velocity rescaling thermostat and the Parrinello–Rahman barostat to maintain the temperature at 300 K and 1 bar. The integration time step was chosen to be 20 fs. To avoid the N terminus aggregation at the entrance of the binding pocket, the backbone of residues 35–44 was restrained during the simulations. Twenty independent coarse-grained simulations were conducted.

### Molecular dynamics simulations
The molecular dynamics simulations were carried out using Gromacs v.2021.4 (ref. 77). The system was constructed using CHARMM-GUI[78–80], with a lipid bilayer composed of 72 cholesterol, 24 1-palmitoyl-2-oleoyl-glycero-3-phosphocholine (POPC), 25 1,2-dipalmitoyl-sn-glycero-3-phosphocholine (DPPC), 12 1-palmitoyl-2-oleoyl-sn-glycero-3-phosphoethanolamine (POPE), 12 1,2-dipalmitoyl-sn-glycero-3-phosphoethanolamine (DPPE), 34 N-palmitoyl-sphingomyelin (PSM) and 18 1,2-dioleoyl-sn-glycero-3-phospho-L-serine (DOPS) to mimic the plasma membrane[81]. The constructed membrane protein system was then solvated and salted with 0.15 M NaCl with extra sodium ions introduced to neutralize the system. Before the production runs, the system was subjected to a 5,000-step restrained energy minimization and six stages of NPT equilibration runs, where the restraints on the protein backbone, side chains and lipids are slowly released. Different from the default CHARMM-GUI setup, we extended the equilibration times to 250 ps, 250 ps, 500 ps, 1 ns, 1 ns and 2 ns. Three copies of a 100 ns production run were performed using the leap-frog integrator with a step size of 2 fs. The van der Waals interaction was gradually switched off between 1.0 and 1.2 nm, while the electrostatic interaction was calculated via the fast smooth particle-mesh Ewald summation with a cutoff at 1.2 nm (ref. 82). The Nosé–Hoover thermostat and the Parrinello–Rahman barostat were used to maintain the system temperature and the system pressure at 303.15 K and 1.0 bar, respectively[83–85]. All bonds involving hydrogen were constrained using LINCS[86]. The cluster analysis was performed using the gromos algorithm with the protein aligned and a ligand r.m.s.d. cutoff of 3 Å (ref. 87). The gromos algorithm counts neighbors based on the r.m.s.d. cutoff. The structure with the most neighbors was taken as the centroid for the first cluster, and all structures of this cluster were then removed from the pool. This procedure was repeated to generate other clusters until the pool was empty.

## Statistics and reproducibility

For split-luciferase-based cAMP reporter assays, data were analyzed using 'log(agonist)' versus 'response' in GraphPad Prism v.8.0. In Figs. 2c,f, 3d and 4d and Extended Data Fig. 4a, data were normalized to the percentage agonist activity of wild-type CXCR3. In Fig. 6e–g, data were normalized to the percentage agonist activity in the presence of 0 nM SCH546738 for CXCR3$^{WT}$, CXCR3$^{V261F}$ and CXCR3$^{A265F}$, respectively. All experiments were repeated independently at least twice with similar results.

## Reporting summary

Further information on research design is available in the Nature Portfolio Reporting Summary linked to this article.

## Data availability

The cryo-EM density maps generated in this study have been deposited in the EMDB under accession codes EMD-34914 (CXCR3–CXCL11–DNG$_i$–scFv16), EMD-34915 (CXCR3–PS372424–DNG$_i$–scFv16), EMD-34916 (CXCR3–VUF11222–DNG$_i$–scFv16) and EMD-34917 (CXCR3$^{kOR}$–SCH546738-Nb6). The associated protein models have been deposited in the PDB under accession codes 8HNK (CXCR3–CXCL11–DNG$_i$–scFv16), 8HNL (CXCR3–PS372424–DNG$_i$–scFv16), 8HNM (CXCR3–VUF11222–DNG$_i$–scFv16) and 8HNN (CXCR3$^{kOR}$–SCH546738-Nb6). Source data are provided with this paper.

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

## Acknowledgements

This work was in part supported by the National Natural Science Foundation of China, Youth Science Fund (project no. 32100963 to H.H.), and Shenzhen Science and Technology Innovation Committee (project no. JCYJ20210324131802008 to H.H.). This work was also supported by the Kobilka Institute of Innovative Drug Discovery and Presidential Fellowship and University Development Fund at the Chinese University of Hong Kong, Shenzhen (H.H., H.J. and Q.C.). Q.P. was supported in part by Presidential Fellowship at the Chinese University of Hong Kong, Shenzhen, and the Ganghong Young Scholar Fund. We also thank the Kobilka Cryo-EM Center at the Chinese University of Hong Kong, Shenzhen, for supporting EM data collection.

## Author contributions

H.J. expressed and purified the complex, prepared the grids, collected and processed the cryo-EM data, built and analyzed the protein atomic model. B.P. accomplished the cAMP assay. A.L. helped with the processing of the cryo-EM data of CXCR3$^{kOR}$–SCH546738-Nb6. Q.C. contributed to data collection. Q.P. and X.W. helped with baculovirus preparation. Y.X. and Y.-C.C. conducted the molecular dynamics simulations. H.J., B.P., Y.-C.C., R.R. and H.H. wrote the manuscript. H.H. coordinated the whole work.

## Competing interests

The authors declare no competing interests.

## Additional information

**Extended data** is available for this paper at https://doi.org/10.1038/s41594-023-01175-5.

**Correspondence and requests for materials** should be addressed to Ying-Chih Chiang, Ruobing Ren or Hongli Hu.

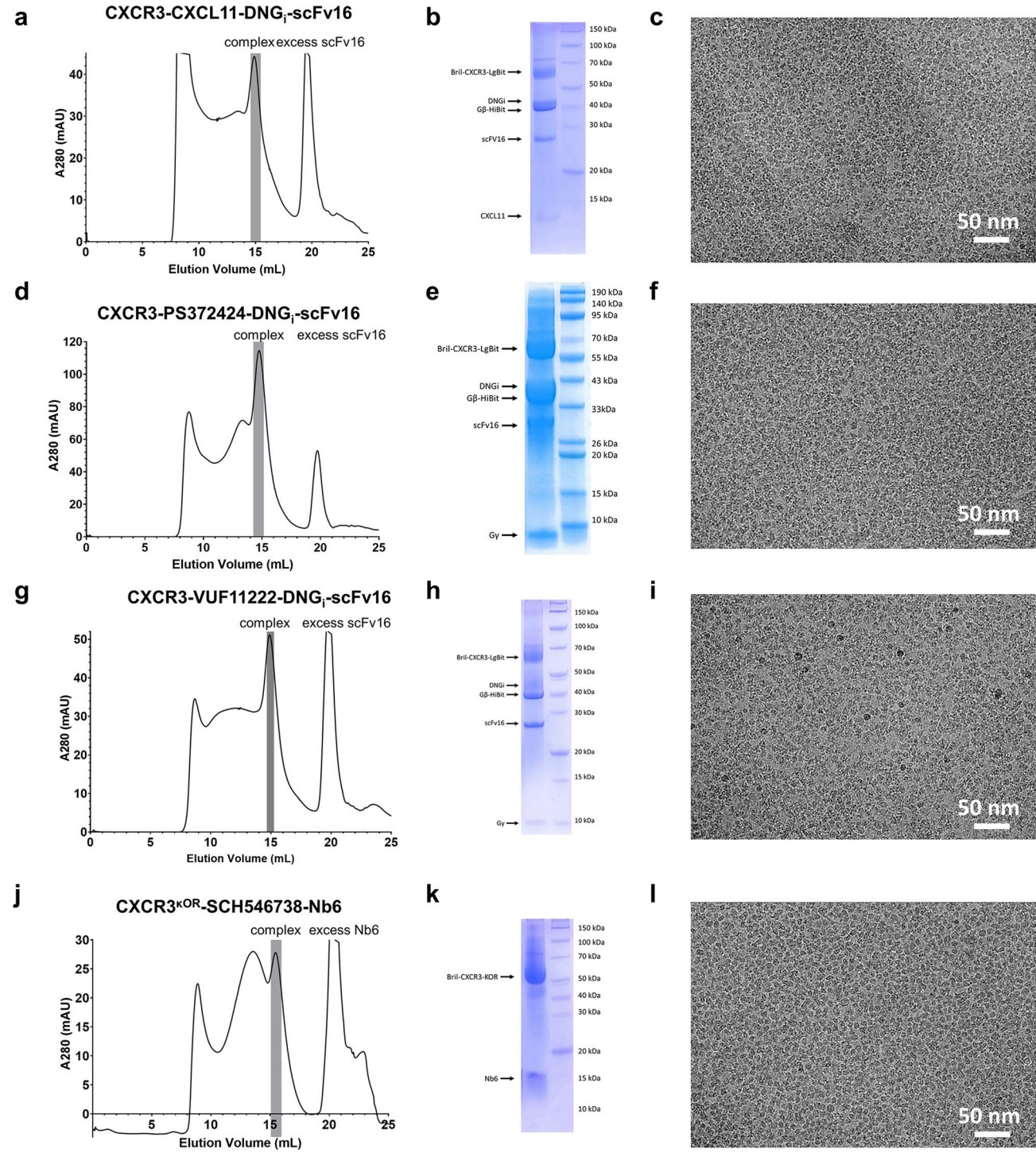

**Extended Data Fig. 1 | Complex preparation and cryo-EM data collection. a** Elution curve of CXCR3-CXCL11-DNG$_i$-scFv16 complex on Superose 6 Increase column. **b** SDS-PAGE of CXCR3-CXCL11-DNG$_i$-scFv16 complex after size-exclusion chromatography. **c** Cryo-EM image of CXCR3-CXCL11-DNG$_i$-scFv16 particles (a representative of 5,500 micrographs). **d** Elution curve of CXCR3-PS372424-DNG$_i$-scFv16 complex on Superose 6 Increase column. **e** SDS-PAGE of CXCR3-PS372424-DNG$_i$-scFv16 complex after size-exclusion chromatography. **f** Cryo-EM image of CXCR3-PS372424-DNG$_i$-scFv16 particles (a representative of 3,185 micrographs). **g** Elution curve of CXCR3-VUF11222-DNG$_i$-scFv16 complex on Superose 6

Increase column. **h** SDS-PAGE of CXCR3-VUF11222-DNG$_i$-scFv16 complex after size-exclusion chromatography. **i** Cryo-EM image of CXCR3-VUF11222-DNG$_i$-scFv16 particles (a representative of 2,891 micrographs). **j** Elution curve of CXCR3$^{kOR}$-SCH546738-Nb6 complex on Superose 6 Increase column. **k** SDS-PAGE of CXCR3$^{kOR}$-SCH546738-Nb6 complex after size-exclusion chromatography. **l** Cryo-EM image of CXCR3$^{kOR}$-SCH546738-Nb6 particles (a representative of 12,944 micrographs). In b, e, h, and k, data shown are representative of two independent experiments.

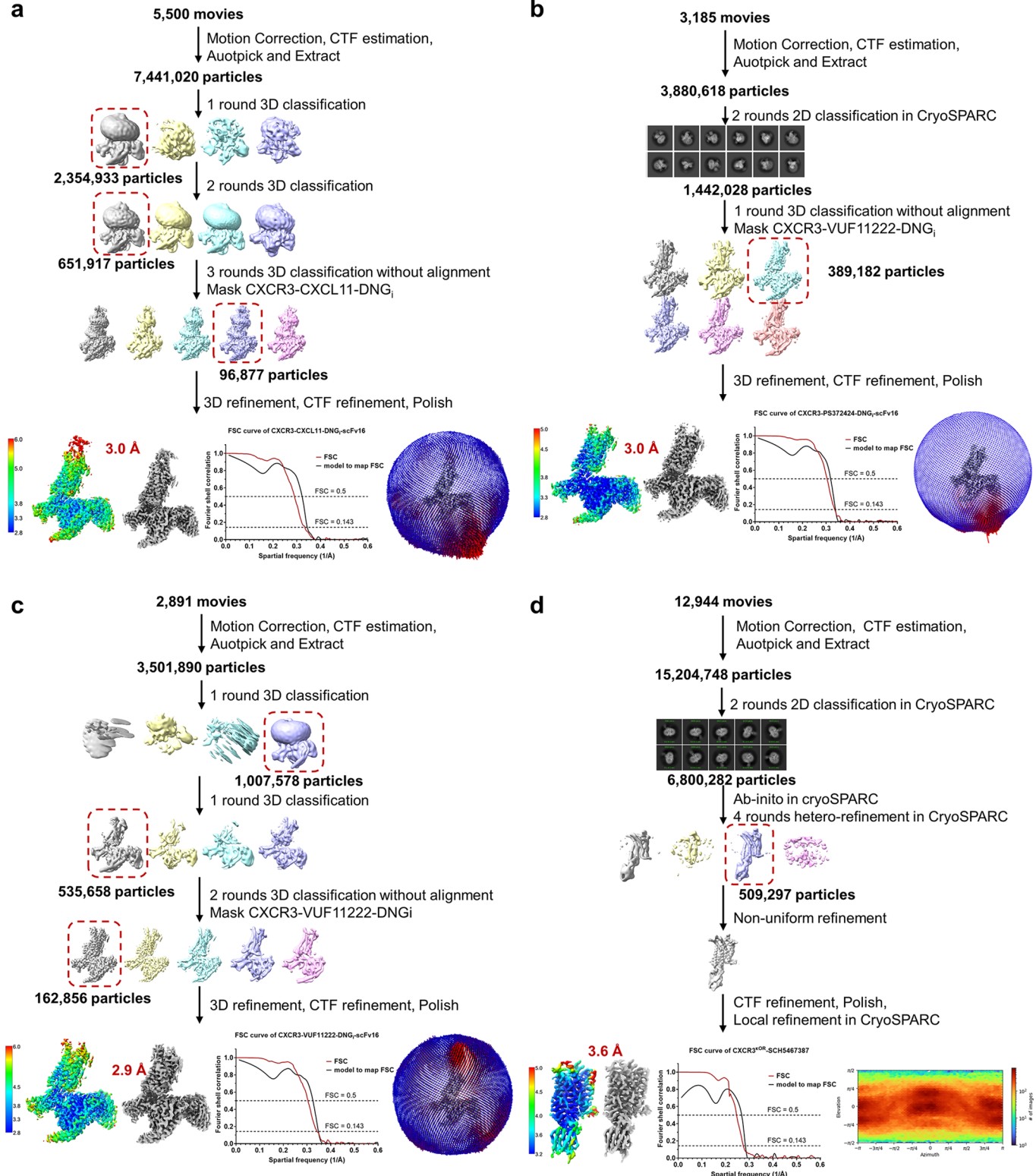

**Extended Data Fig. 2 | Cryo-EM data processing. a** Cryo-EM data processing of CXCR3-CXCL11-DNG$_i$-scFv16. The local resolution map and the FSC curve are presented as well. **b** Cryo-EM data processing of CXCR3-PS372424-DNG$_i$-scFv16. The local resolution map and the FSC curve are presented as well. **c** Cryo-EM data processing of CXCR3-VUF11222-DNG$_i$-scFv16. The local resolution map and the FSC curve are presented as well. **d** Cryo-EM data processing of CXCR3$^{\kappa OR}$-SCH546738-Nb6. The local resolution map and the FSC curve are presented as well.

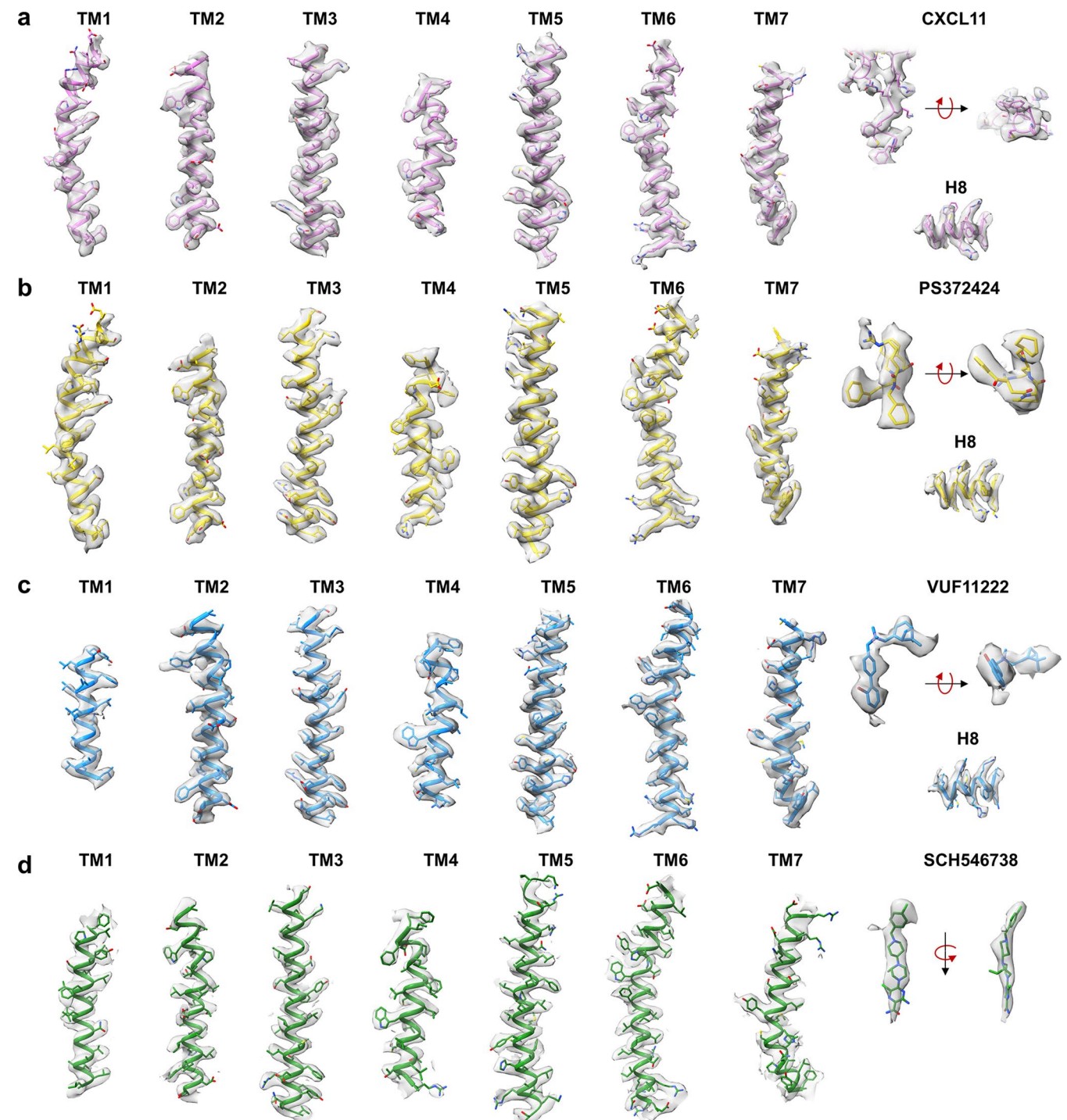

**Extended Data Fig. 3 | Local density presentation. a** Density maps of the transmembrane helixes and CXCL11 in the structure of CXCR3-CXCL11-DNG$_i$-scFv16. **b** Density maps of the transmembrane helixes and PS372424 in the structure of CXCR3- PS372424 -DNG$_i$-scFv16. **c** Density maps of the transmembrane helixes and VUF11222 in the structure of CXCR3- VUF11222 -DNG$_i$-scFv16. **d** Density maps of the transmembrane helixes and SCH546738 in the structure of CXCR3$^{κOR}$- SCH546738-Nb6.

**a**

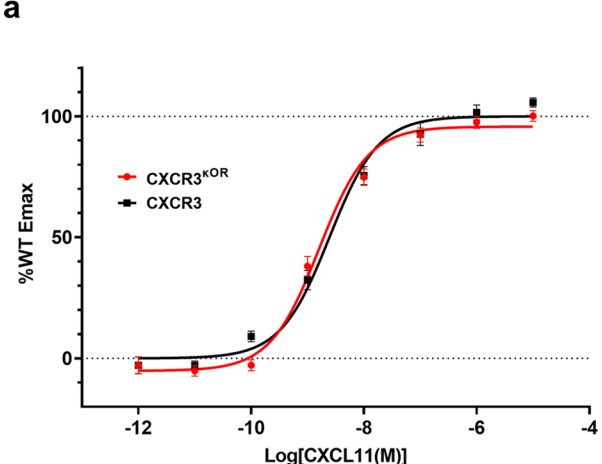
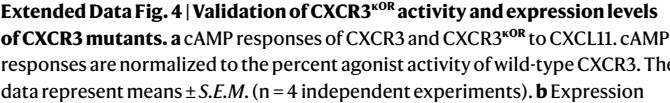

**b**

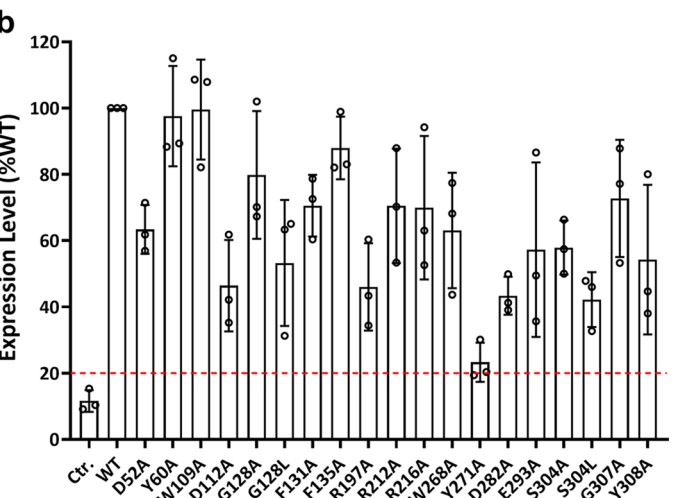

**Extended Data Fig. 4 | Validation of CXCR3$^{KOR}$ activity and expression levels of CXCR3 mutants. a** cAMP responses of CXCR3 and CXCR3$^{KOR}$ to CXCL11. cAMP responses are normalized to the percent agonist activity of wild-type CXCR3. The data represent means $\pm$ *S.E.M.* (n = 4 independent experiments). **b** Expression levels of CXCR3 mutants determined by quantitative flow cytometry. The Red dashed line indicates a 20% expression level of wild-type CXCR3. The data are shown as means $\pm$ *S.E.M.* of three independent experiments.

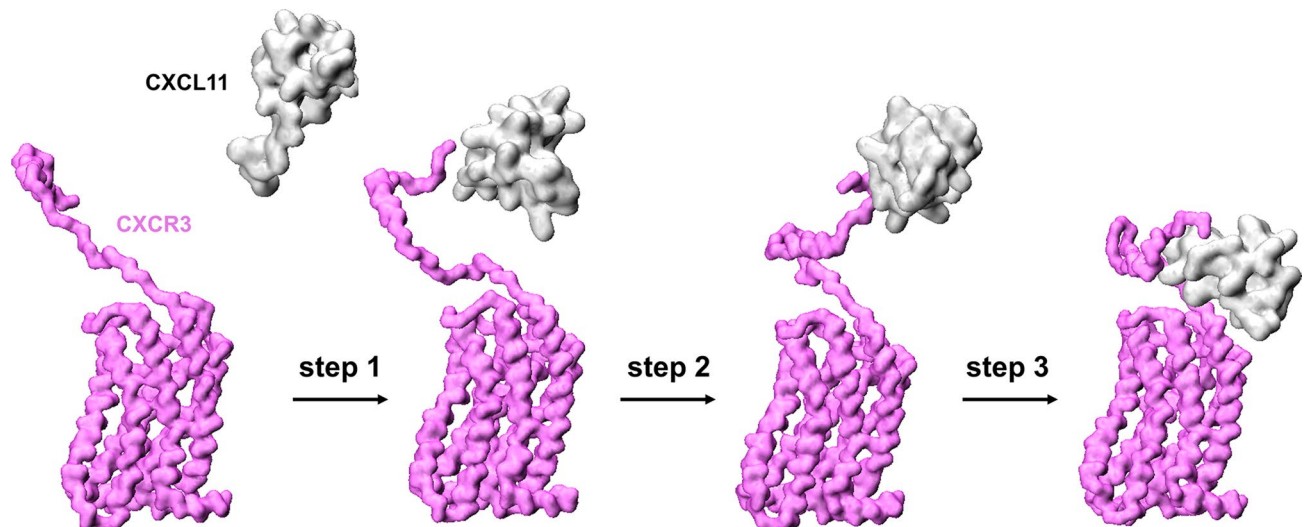

**Extended Data Fig. 5 | The dynamic process of CXCL11 recruitment revealed by coarse-grained molecular dynamic simulation.** The backbone of CXCR3 is shown as surface and colored violet, CXCL11 is shown as surface and colored gray.

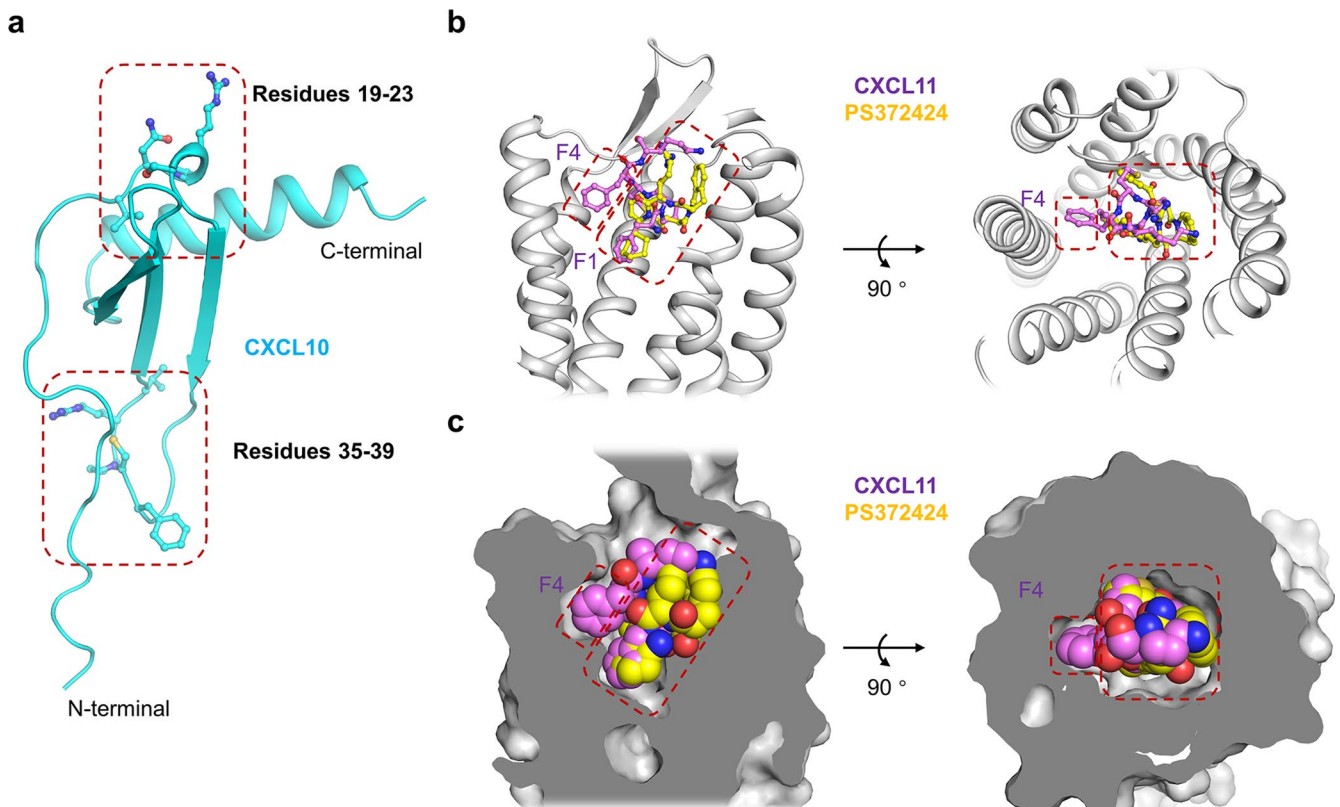

**Extended Data Fig. 6 | The binding pattern of PS372424 is similar to CXCL11. a** The model of CXCL10 predicted by AlphaFold2. CXCL10 is shown as cartoon model and colored cyan, residues 19-23 and 35-39 are shown as sticks. **b** Superposition of the binding site of PS372424 and CXCL11. The receptor is shown as cartoon and colored gray. PS372424 and the N-terminal pentapeptide of CXCL11 are shown as sticks and colored yellow and violet, respectively. **c** Superposition of the binding pocket of PS372424 and CXCL11. The receptor is shown as surface and colored gray. PS372424 and the N-terminal pentapeptide of CXCL11 are shown as spheres and colored yellow and violet, respectively.

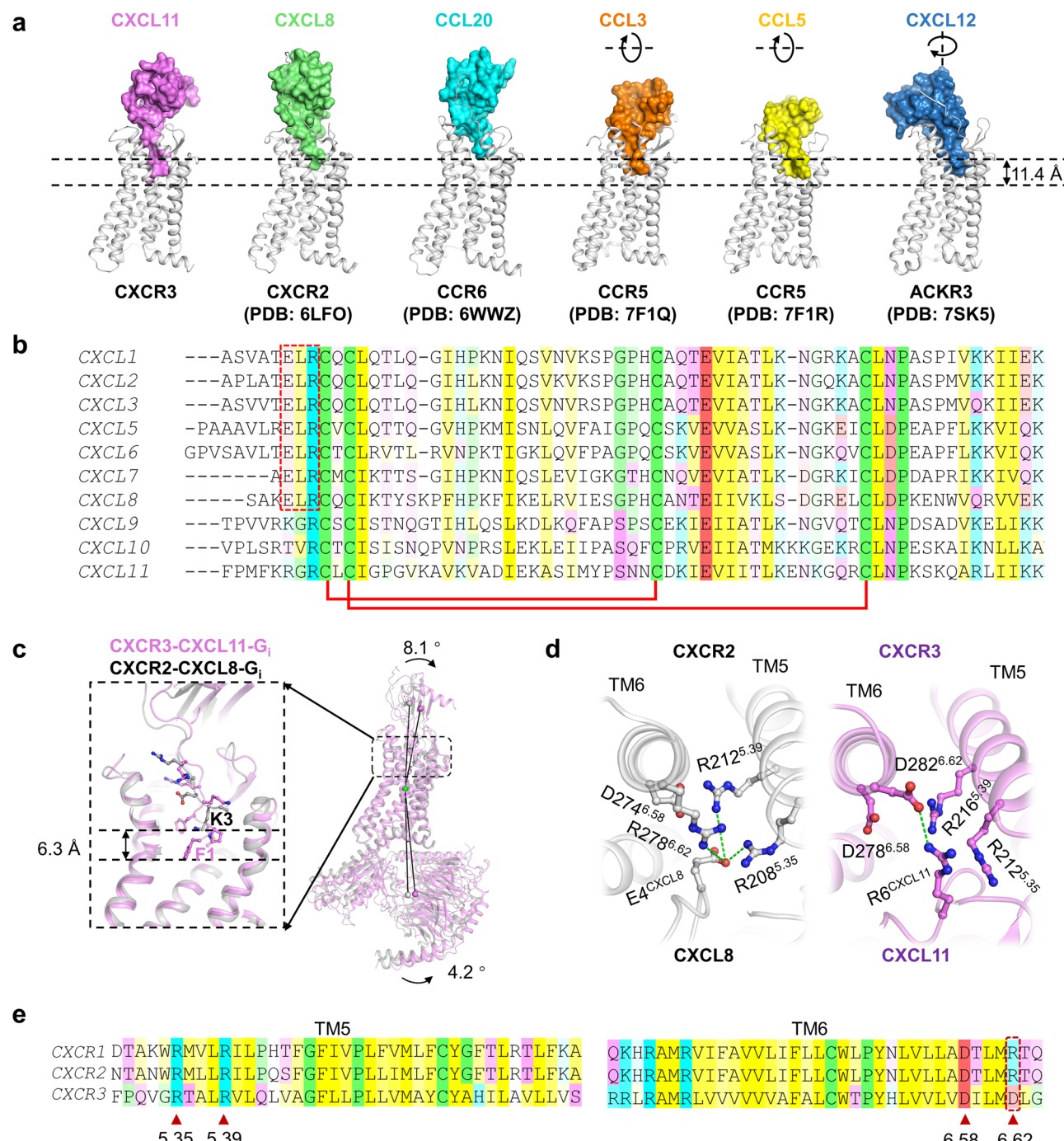

**Extended Data Fig. 7 | Structural basis for the chemokine selectivity of CXCR2 and CXCR3. a** Comparison of the insertion depth of CXCL11 (violet), CXCL8 (green), CCL20 (cyan), CCL3 (orange), CCL5 (yellow), and CXCL12 (blue). The receptors are shown as cartoon and colored gray, while the chemokines are shown in surface model. The pseudo rotation axis and orientation for CCL3, CCL5, and CXCL12 compared to CXCL11 are shown. **b** Sequence alignment of CXCL1-3 and 5-11. The disulfide bonds are indicated as red line links. The ELR motif in CXCL1-3 and 5-8 are indicated by a dotted box. **c** Comparison of CXCR3-CXCL11-G$_i$ (violet) and CXCR2-CXCL8-G$_i$ (gray). The angles are measured using the mass centers of the receptors, the chemokines, and the G$_i$ complexes as reference points. **d** Residues in CXCR2 and CXCR3 that are involved in the binding of E4$^{CXCL8}$ and R6$^{CXCL11}$. The structure of CXCR2 and CXCR3 are shown as cartoon and colored gray and violet, respectively. Residues involved in interactions are shown as sticks, and interactions are indicated by green dashes. **e** Sequence alignment of TM5 and TM6 in CXCR1, CXCR2 and CXCR3. Key residues in TM5 and TM6 are indicated by red-colored triangles. In **b** and **e**, positively charged residues, negatively charged residues, non-charged hydrophilic residues, and hydrophobic residues are colored cyan, red, violet, and yellow, respectively. Residues cystine, proline, and glycine are colored green. The value of conservation visibility is set to 30% in JalView.

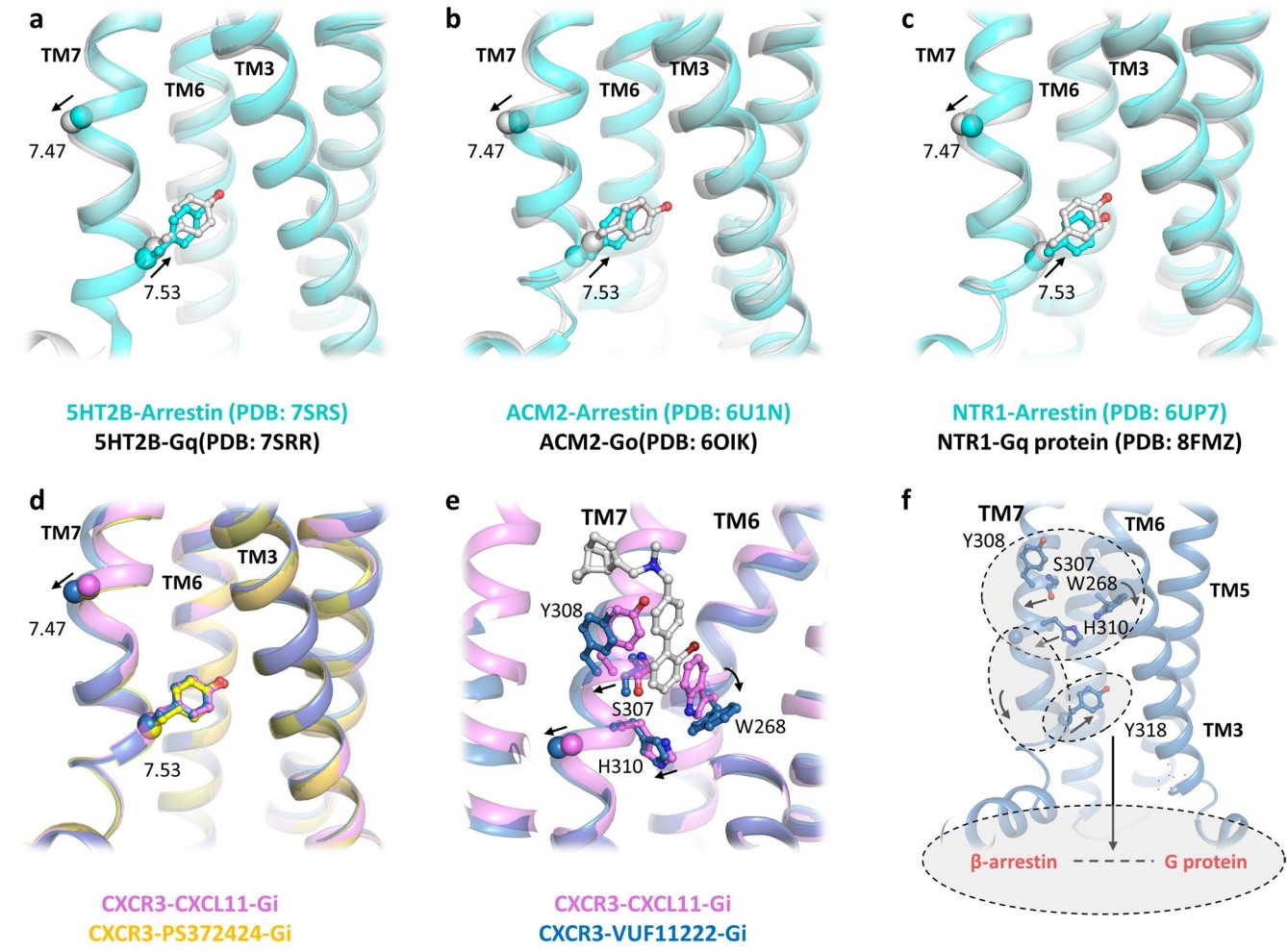

**5HT2B-Arrestin (PDB: 7SRS)**
**5HT2B-Gq(PDB: 7SRR)**

**ACM2-Arrestin (PDB: 6U1N)**
**ACM2-Go(PDB: 6OIK)**

**NTR1-Arrestin (PDB: 6UP7)**
**NTR1-Gq protein (PDB: 8FMZ)**

**CXCR3-CXCL11-Gi**
**CXCR3-PS372424-Gi**
**CXCR3-VUF11222-Gi**

**CXCR3-CXCL11-Gi**
**CXCR3-VUF11222-Gi**

**Extended Data Fig. 8 | A proposed mechanism for the bias signaling of CXCR3. a** Superposition of the β-arrestin-coupled and G protein-coupled 5HT2B. **b** Superposition of the β-arrestin-coupled and G protein-coupled ACM2. **c** Superposition of the β-arrestin-coupled and G protein-coupled NTR1. In **a-c**, the receptors are shown as cartoon, and colored cyan for β-arrestin-coupled receptor and gray for G protein-coupled receptor. The Cα atoms of 7.47 and 7.53 are shown as spheres, and the side chains of 7.53 are shown as sticks. The tilt in TM7$^{7.47-7.53}$ is indicated by black arrows. **d** Superposition of Gi-coupled CXCR3 activated by CXCL11 (violet), PS372424 (yellow), and VUF11222 (blue). **e** The tilt of TM7$^{7.47-7.53}$ initiated by the insertion of VUF11222. **f** A proposed model for the mechanism for the biased signaling of CXCR3.

**Extended Data Table 1 | The potencies of CXCR3 ligands with the receptor or its mutants**

|  | Mutants | logEC50 ± Std. error | EC$_{50}$ (M) | N Value | Fold |
|---|---|---|---|---|---|
| **CXCL11** | **WT** | -8.65 ± 0.05 | $2.25 \times 10^{-9}$ | 4 | 1.0 |
|  | **D52A** | -6.81 ± 0.06 | $1.55 \times 10^{-7}$ | 3 | 69 |
|  | **W109A** | -6.73 ± 0.15 | $1.86 \times 10^{-7}$ | 3 | 83 |
|  | **F131A** | -7.62 ± 0.07 | $2.42 \times 10^{-8}$ | 3 | 11 |
|  | **F135A** | -6.52 ± 0.09 | $3.05 \times 10^{-7}$ | 3 | 136 |
|  | **R197A** | -7.14 ± 0.06 | $7.20 \times 10^{-8}$ | 3 | 32 |
|  | **R212A** | -7.20 ± 0.07 | $6.27 \times 10^{-8}$ | 3 | 28 |
|  | **Y271A** | -7.92 ± 0.09 | $1.19 \times 10^{-8}$ | 3 | 5.3 |
|  | **D282A** | -7.71 ± 0.13 | $1.94 \times 10^{-8}$ | 3 | 8.6 |
|  | **E293A** | -6.72 ± 0.08 | $1.92 \times 10^{-7}$ | 3 | 85 |
|  | **S304L** | -7.46 ± 0.11 | $3.48 \times 10^{-8}$ | 3 | 15 |
|  | **Y308A** | -8.02 ± 0.09 | $9.46 \times 10^{-9}$ | 3 | 4.2 |
| **PS372424** | **WT** | -8.27 ± 0.08 | $5.36 \times 10^{-9}$ | 3 | 1.0 |
|  | **Y60A** | -7.46 ± 0.20 | $3.46 \times 10^{-8}$ | 3 | 6.5 |
|  | **W109A** | -7.28 ± 0.11 | $5.25 \times 10^{-8}$ | 3 | 9.8 |
|  | **F131A** | -7.42 ± 0.07 | $3.77 \times 10^{-8}$ | 3 | 7.0 |
|  | **F135A** | -6.48 ± 0.07 | $3.34 \times 10^{-7}$ | 3 | 62 |
|  | **Y271A** | -8.08 ± 0.09 | $8.32 \times 10^{-9}$ | 3 | 1.6 |
|  | **Y308A** | -6.50 ± 0.06 | $3.14 \times 10^{-7}$ | 3 | 59 |
| **VUF11222** | **WT** | -7.25 ± 0.12 | $5.59 \times 10^{-8}$ | 4 | 1.0 |
|  | **Y60A** | -7.27 ± 0.53 | $5.42 \times 10^{-8}$ | 4 | 1.0 |
|  | **W109A** | -7.64 ± 0.10 | $2.28 \times 10^{-8}$ | 4 | 0.4 |
|  | **F131A** | -6.23 ± 0.35 | $5.85 \times 10^{-7}$ | 4 | 10 |
|  | **F135A** | -5.65 ± 0.35 | $2.27 \times 10^{-6}$ | 4 | 41 |
|  | **W268A** | -7.15 ± 0.23 | $7.09 \times 10^{-8}$ | 4 | 1.3 |
|  | **Y271A** | -6.76 ± 0.15 | $1.76 \times 10^{-7}$ | 4 | 3.1 |
|  | **S304L** | -7.21 ± 0.08 | $6.22 \times 10^{-8}$ | 4 | 1.1 |
|  | **G307A** | -5.37 ± 0.40 | $4.31 \times 10^{-6}$ | 4 | 77 |
|  | **Y308A** | -7.61 ± 0.09 | $2.47 \times 10^{-8}$ | 4 | 0.4 |

**Extended Data Table 2 | The potencies of CXCL11 with the receptor or its mutants in the present of different concentrations of SCH546738**

| | SCH546738 (nM) | logEC50 ± Std. error | $EC_{50}$ (M) | N Value |
|---|---|---|---|---|
| **CXCR3$^{WT}$** | **0** | -8.37 ± 0.07 | $4.26 \times 10^{-9}$ | 6 |
| | **10** | -7.57 ± 0.12 | $2.70 \times 10^{-8}$ | 6 |
| | **100** | -7.28 ± 0.06 | $5.31 \times 10^{-8}$ | 6 |
| | **1000** | -6.94 ± 0.07 | $1.16 \times 10^{-7}$ | 6 |
| | **5000** | -6.74 ± 0.09 | $1.84 \times 10^{-7}$ | 6 |
| | **10000** | -5.96 ± 0.07 | $1.10 \times 10^{-6}$ | 6 |
| **CXCR3$^{V261F}$** | **0** | -8.48 ± 0.06 | $3.33 \times 10^{-9}$ | 6 |
| | **10** | -7.86 ± 0.06 | $1.39 \times 10^{-8}$ | 6 |
| | **100** | -8.83 ± 0.12 | $1.49 \times 10^{-9}$ | 6 |
| | **1000** | -8.20 ± 0.08 | $6.38 \times 10^{-9}$ | 6 |
| | **5000** | -8.45 ± 0.10 | $3.57 \times 10^{-9}$ | 6 |
| | **10000** | -8.12 ± 0.06 | $7.58 \times 10^{-9}$ | 6 |
| **CXCR3$^{A265F}$** | **0** | -7.47 ± 0.08 | $3.42 \times 10^{-8}$ | 6 |
| | **10** | -7.18 ± 0.07 | $6.69 \times 10^{-8}$ | 6 |
| | **100** | -7.02 ± 0.08 | $9.55 \times 10^{-8}$ | 6 |
| | **1000** | -7.47 ± 0.06 | $3.36 \times 10^{-8}$ | 6 |
| | **5000** | -7.39 ± 0.06 | $4.05 \times 10^{-8}$ | 6 |
| | **10000** | -6.91 ± 0.05 | $1.24 \times 10^{-7}$ | 6 |

# Reporting Summary

## Statistics

For all statistical analyses, confirm that the following items are present in the figure legend, table legend, main text, or Methods section.

| n/a | Confirmed | |
|---|---|---|
| ☐ | ☒ | The exact sample size (*n*) for each experimental group/condition, given as a discrete number and unit of measurement |
| ☐ | ☒ | A statement on whether measurements were taken from distinct samples or whether the same sample was measured repeatedly |
| ☒ | ☐ | The statistical test(s) used AND whether they are one- or two-sided *Only common tests should be described solely by name; describe more complex techniques in the Methods section.* |
| ☒ | ☐ | A description of all covariates tested |
| ☒ | ☐ | A description of any assumptions or corrections, such as tests of normality and adjustment for multiple comparisons |
| ☐ | ☒ | A full description of the statistical parameters including central tendency (e.g. means) or other basic estimates (e.g. regression coefficient) AND variation (e.g. standard deviation) or associated estimates of uncertainty (e.g. confidence intervals) |
| ☒ | ☐ | For null hypothesis testing, the test statistic (e.g. *F*, *t*, *r*) with confidence intervals, effect sizes, degrees of freedom and *P* value noted *Give P values as exact values whenever suitable.* |
| ☒ | ☐ | For Bayesian analysis, information on the choice of priors and Markov chain Monte Carlo settings |
| ☒ | ☐ | For hierarchical and complex designs, identification of the appropriate level for tests and full reporting of outcomes |
| ☒ | ☐ | Estimates of effect sizes (e.g. Cohen's *d*, Pearson's *r*), indicating how they were calculated |

*Our web collection on statistics for biologists contains articles on many of the points above.*

## Software and code

Policy information about availability of computer code

| Data collection | SerialEM 3.8 |
|---|---|
| Data analysis | MotionCor2, Gctf 1.18, RELION 3.1, CryoSPARC 3.3.1, UCSF ChimeraX 1.4, Coot 0.9.8.1, PyMOL 2.0, Phenix 1.20.1, CCP-EM 1.6.0, AceDRG 246, LocScale, GraphPad Prism 8.0, CHARMM-GUI 3.8, Gromacs 2021.4. |

For manuscripts utilizing custom algorithms or software that are central to the research but not yet described in published literature, software must be made available to editors and reviewers. We strongly encourage code deposition in a community repository (e.g. GitHub). See the Nature Portfolio guidelines for submitting code & software for further information.

## Data

Policy information about availability of data

All manuscripts must include a data availability statement. This statement should provide the following information, where applicable:

- Accession codes, unique identifiers, or web links for publicly available datasets
- A description of any restrictions on data availability
- For clinical datasets or third party data, please ensure that the statement adheres to our policy

The cryo-EM density maps generated in this study have been deposited in Electron Microscopy Data Bank under accession codes: EMD-34914 (CXCR3-CXCL11-DNGi-scFv16), EMD-34915 (CXCR3-PS372424-DNGi-scFv16), EMD-34916 (CXCR3-VUF11222-DNGi-scFv16), and EMD-34917 (CXCR3κOR-SCH546738-Nb6). The

associated protein models have been deposited in the Protein Data Bank under accession codes: 8HNK (CXCR3-CXCL11-DNGi-scFv16), 8HNL (CXCR3-PS372424-DNGi-scFv16), 8HNM (CXCR3-VUF11222-DNGi-scFv16), and 8HNN (CXCR3κOR-SCH546738-Nb6). Source data are provided with this paper.

## Research involving human participants, their data, or biological material

Policy information about studies with human participants or human data. See also policy information about sex, gender (identity/presentation), and sexual orientation and race, ethnicity and racism.

| | |
|---|---|
| Reporting on sex and gender | N/A |
| Reporting on race, ethnicity, or other socially relevant groupings | N/A |
| Population characteristics | N/A |
| Recruitment | N/A |
| Ethics oversight | N/A |

Note that full information on the approval of the study protocol must also be provided in the manuscript.

## Field-specific reporting

Please select the one below that is the best fit for your research. If you are not sure, read the appropriate sections before making your selection.

☒ Life sciences ☐ Behavioural & social sciences ☐ Ecological, evolutionary & environmental sciences

For a reference copy of the document with all sections, see nature.com/documents/nr-reporting-summary-flat.pdf

## Life sciences study design

All studies must disclose on these points even when the disclosure is negative.

| | |
|---|---|
| Sample size | No sample calculation was performed for the cryo-EM analysis. The sample size of cryo-EM micrographs was determined by the availability of the microscope time. The cryo-EM micrographs collected are: 5500 (CXCR3-CXCL11-DNGi-scFv16), 3185 (CXCR3-PS372424-DNGi-scFv16), 2891 (CXCR3-VUF11222-DNGi-scFv16), and 12944 (CXCR3κOR-SCH546738-Nb6). The sample size of particles used in the final reconstruction was determined by reported resolution and the quality of the density map. The final particles used for cryo-EM reconstruction are: 96877 (CXCR3-CXCL11-DNGi-scFv16), 389182 (CXCR3-PS372424-DNGi-scFv16), 162856 (CXCR3-VUF11222-DNGi-scFv16), and 509297 (CXCR3κOR-SCH546738-Nb6).<br>For cAMP assay, 3 to 6 independent experiments were performed as indicated in the figure legends. |
| Data exclusions | During data processing, the particles with bad alignment were excluded from the final reconstruction based on the 2D averages or 3D classification map as implemented in RELION and CryoSPARC. |
| Replication | For cryo-EM, no replication studies were attempted. The primary data is cryo-EM structures that are calculated according to standard procedures and do not need replicates. A replication of the cryo-EM data collection with the same sample size was not economically justifiable and the time cost was high.<br>For cAMP assay, each experiment was replicated at least twice on separate occasions. |
| Randomization | Randomization is irrelevant for our study because no grouping was needed. For single-particle cryo-EM analysis, particles are divided into two random sets for map reconstruction and estimation of the resolution via the Fourier Shell Correlation (FSC) method. |
| Blinding | Blinding is irrelevant to our study because no allocation into experimental groups was needed. |

## Reporting for specific materials, systems and methods

We require information from authors about some types of materials, experimental systems and methods used in many studies. Here, indicate whether each material, system or method listed is relevant to your study. If you are not sure if a list item applies to your research, read the appropriate section before selecting a response.

## Materials & experimental systems

| n/a | Involved in the study |
|---|---|
| ☐ | ☒ Antibodies |
| ☐ | ☒ Eukaryotic cell lines |
| ☒ | ☐ Palaeontology and archaeology |
| ☒ | ☐ Animals and other organisms |
| ☒ | ☐ Clinical data |
| ☒ | ☐ Dual use research of concern |
| ☒ | ☐ Plants |

## Methods

| n/a | Involved in the study |
|---|---|
| ☒ | ☐ ChIP-seq |
| ☒ | ☐ Flow cytometry |
| ☒ | ☐ MRI-based neuroimaging |

## Antibodies

| | |
|---|---|
| Antibodies used | Anti Flag-Tag Mouse Monoclonal Antibody (CWBIO, Cat# CW0287), clone name F-tag-01<br>Alexa Fluor 488-labeled Goat Anti-Mouse IgG (Beyotime, Cat# A0428) |
| Validation | Anti Flag-Tag Mouse Monoclonal Antibody (CWBIO, Cat# CW0287)<br>Validation statement from manufacturer's website: The antibody is highly specific to recognize the Flag tag at the C-terminal or N-terminal of the recombinant protein without being affected by neighboring amino acids, and can be used to detect Flag-tag fusion proteins expressed by various expression vectors. Clone antibody type: rat monoclonal antibody. IgG number: F-tag-01. Immunogen: synthetic peptide (DYKDDDDK). Antibody concentration: 0.5 mg/ml. Application: WB (1:500-5000), IP (1:50-200), IF/ICC, ELISA.<br>Alexa Fluor 488-labeled Goat Anti-Mouse IgG (Beyotime, Cat# A0428)<br>Validation statement from manufacturer's website: This Alexa Fluor 488-labeled Goat Anti-Mouse IgG(H+L) could be used for immunofluorescence staining. The antibody is produced by immuning goats with purified mouse IgG, and has little binding ability to human IgG, horse IgG, bovine IgG(bovine IgG), rabbit IgG and pig IgG. It is especially suitable for fluorescent staining experiment which requires high specificity of secondary antibody species. This Alexa Fluor 488-labeled Goat anti-mouse IgG(H+L) has a recommended dilution ratio of 1:500 for immunofluorescence staining. |

## Eukaryotic cell lines

Policy information about cell lines and Sex and Gender in Research

| | |
|---|---|
| Cell line source(s) | Sf9 (Thermo Fisher Scientific, Cat# 11496015)<br>HEK293T (ATCC, Cat# CRL-11268) |
| Authentication | None of the cell lines used were authenticated. |
| Mycoplasma contamination | Cell lines were not tested for mycoplasma contamination. |
| Commonly misidentified lines<br>(See ICLAC register) | No commonly misidentified cell lines were used in the study. |

