## [Peer Review File · Nature Structural & Molecular Biology]

Peer Review Information

Manuscript Title: Structural insights into the activation and inhibition of CXC chemokine receptor 3

Corresponding author name(s): Hongli Hu, Ruobing Ren, Ying-Chih Chiang

Reviewer Comments & Decisions:

Decision Letter, initial version:
--

Message: 27th May 2023

Dear Dr. Hu,

Thank you again for submitting your manuscript "Structural insights into the activation and inhibition of CXC chemokine receptor 3". I apologize for the delay in responding, which resulted from the difficulty in obtaining suitable referee reports. Nevertheless, we now have comments (below) from the 2 reviewers who evaluated your paper. In light of those reports, we remain interested in your study and would like to see your response to the comments of the referees, in the form of a revised manuscript.

You will see that while the reviewers find the results interesting, they raise several concerns which will need to be addressed in a revision. Specifically, we agree with reviewer #1 that further mechanistic investigation would strengthen the study, particularly exploring receptor aspects of receptor activation and downstream signalling, if feasible. Furthermore, we would request biochemical and functional characterisation of the chimeric protein construct, in line with reviewer's #1 comments, and comparison of expression levels between WT and mutant proteins as pointed out by reviewer #2. To further strengthen conclusions pertaining to the allosteric regulation, we would encourage further validation of these experiments.

Please be sure to address/respond to all concerns of the referees in full in a point-by-point response and highlight all changes in the revised manuscript text file. If you have comments that are intended for editors only, please include those in a separate cover letter.

We expect to see your revised manuscript within 10 weeks. If you cannot send it within

this time, please contact us to discuss an extension; we would still consider your revision, provided that no similar work has been accepted for publication at NSMB or published elsewhere.

Reporting Summary:

Please note that all key data shown in the main figures as cropped gels or blots should be presented in uncropped form, with molecular weight markers. These data can be aggregated into a single supplementary figure item. While these data can be displayed in a relatively informal style, they must refer back to the relevant figures. These data should be submitted with the final revision, as source data, prior to acceptance, but you may want to start putting it together at this point.

SOURCE DATA: we request authors to provide, in tabular form, the data underlying the graphical representations used in figures. This is to further increase transparency in data reporting, as detailed in this editorial (<http://www.nature.com/nsmb/journal/v22/n10/full/nsmb.3110.html>). Spreadsheets can be submitted in excel format. Only one (1) file per figure is permitted; thus, for multi-paneled figures, the source data for each panel should be clearly labeled in the Excel file; alternately the data can be provided as multiple, clearly labeled sheets in an Excel file.

When submitting files, the title field should indicate which figure the source data pertains to. We encourage our authors to provide source data at the revision stage, so that they are part of the peer-review process.

Data availability: this journal strongly supports public availability of data. All data used in accepted papers should be available via a public data repository, or alternatively, as Supplementary Information. If data can only be shared on request, please explain why in your Data Availability Statement, and also in the correspondence with your editor. Please note that for some data types, deposition in a public repository is mandatory - more information on our data deposition policies and available repositories can be found below: <https://www.nature.com/nature-research/editorial-policies/reporting-standards#availability-of-data>

[redacted]

Sincerely,

Katarzyna Ciazynska
(she/her)
Associate Editor
Nature Structural & Molecular Biology
<https://orcid.org/0000-0002-9899-2428>

Referee expertise:

Referee #1: structural biology, GPCRs

Referee #2: biochemistry, GPCRs

Reviewers' Comments:

Reviewer #1:

Remarks to the Author:

This study provides a comprehensive analysis of four CXCR3 structures bound to various ligands, including an endogenous chemokine, two small molecule agonists, and an antagonist. The authors confirmed the binding interfaces of each agonist on CXCR3 using mutagenesis and discovered an allosteric binding site in the antagonist-bound structure that had not been previously identified in class A GPCRs. While these findings offer valuable insights into receptor-ligand interaction, a more in-depth exploration of the mechanisms beyond the binding interface and the newly discovered allosteric binding sites would further enhance the study's impact. To achieve this, it is recommended that the manuscript discusses the impact of these different binding modes on receptor activation and signaling, as well as their potential applications in future drug development efforts.

Major points:

1) Page 2, line 22 "Structural analysis reveals that PS372424 shares a similar orthosteric binding pocket with the N terminus of CXCL11, while VUF11222 buries deeper and activates the receptor in a different way."

The authors concluded that CXCL11 and PS372424 have a similar binding pocket, while VUF11222 binds deeply, resulting in a distinct activation mechanism for the receptor. While the structures do show conformational changes in the extracellular domain to accommodate different ligands, the intracellular sites appear to be highly similar across all three structures. I would appreciate further clarification on the statement that VUF11222 activates the receptor in a distinct manner. I am also interested in learning more about the downstream signaling regulation properties of the three agonists discussed in the paper. I am curious about whether they are neutral or biased agonists, whether they have similar affinity to the receptor and whether the structures provide a possible explanation for their unique properties.

2) In order to obtain the antagonist-bound structure, the authors generated a chimeric protein that involved swapping the ICL3 of CXCR3 with the ICL3 from the kappa-opioid receptor. However, the paper lacks any biochemical data that functionally characterizes

this chimeric protein. Furthermore, there is no mutagenesis data to verify the observed allosteric binding site for the antagonist in the wild-type protein, as was done for the agonists in cell-based assays. These omissions should be addressed experimentally.

3) Page 4, line 99 "The interactions between the proximal N-terminus (chemokine recognition site 1, CRS1) of the receptor and the core domain (chemokine site 1, CS1) of chemokine were reported to be weak and reversible."

Further clarification is needed on what is meant by "reversible" in this context. Please provide a citation to the original paper that supports this claim, rather than relying on a review paper.

4) Page 5, line 115 "Mutation of Trp1092.60, Phe1313.32, Phe1353.36, Tyr2716.51, Ser3047.39, and Tyr3087.43 results in reduced CXCL11-induced signaling response (Fig. 2c, Extended Data Fig. 4, Extended Data Table 2), confirming the essential roles of these residues in CXCL11 binding and receptor activation."

The data presented in Figure 2c suggests that the mutants exhibit the same maximum signal as the wild-type protein, indicating that there is no change in CXCL11-induced signaling responses, but potentially reduced binding affinity. The mutations made on the receptor are helpful in validating the binding interface between the receptor and CXCL11. However, to gain a more complete understanding of the observed interaction, it would be useful to investigate the role of the residues on the chemokine. Therefore, it would be worth summarizing existing literature on how mutations in CXCL11 affect CXCR3 signaling and compare them to the structural observations.

5) The numbering of residues in CXCL11 and receptor in the structure of SCH546738 presented in the provided PDB files do not match the numbering described in the manuscript. For instance, residue E40 in the manuscript is labeled as E61 in the provided PDB file. This discrepancy can be quite confusing and needs to be addressed by correcting the numbering in the PDB file or vice versa.

6) P6, line 145 "In a previous study, three-dimensional alignments of PS372424 to pentapeptides of CXCL10 suggest that PS372424 may mimic the residues 35-39 or 19-23 of CXCL1036." And the conclusion from this paragraph: "we suggest the binding pattern of PS372424 is similar to CXCL11 rather than CXCL10."

The paper cited in the manuscript clearly states that "We emphasize that there is no direct evidence that any small molecule agonist binds to CXCR3 in the same manner as any portion of IP-10. Rather we present this example to show how three-dimensional alignment can be used in a nonstandard manner to generate reasonable testable hypotheses" The paper primarily focuses on the geometry of PS372424 and suggests that its 3D conformation might be similar to either 35-39 or 19-23 of CXCL10. In fact, the structure of PS372424 in your model aligns well with one of their predictions. Therefore, it may not be appropriate to conclude that "the binding pattern of PS372424 is similar to CXCL11 rather than CXCL10" based solely on the cited paper.

7) Page 7, line 184 "Compared with the VUF11222 coupled receptor, a kink in TM1 could be observed in the CXCL11 coupled receptor, making the N-terminus of TM1 bend toward TM7 (Fig.5c). "

In the structure of VUF11222, there is disorder in part of TM1 and the N-terminus when compared to the CXCL11 and PS372424 structures. It is unclear what may have caused this disorder, particularly since the N-terminus is tethered to TM7 by a disulfide bond in the other two structures.

8) Page 9, line 254 "As no structure of chemokine receptors activated by small molecule agonists is available at present, the structures of CXCR3 coupled with small molecule agonists PS372424 and VUF11222 provide an opportunity for comparing the binding pattern of small molecule agonists with chemokines."

This statement is incorrect, and it would be appropriate to refer to a study that demonstrates the structure of an atypical chemokine receptor (ACKR3) bound to a small molecule agonist (PMID: 35857509). The author should cite this paper and make essential revisions to their comment. It may also be advantageous for the author to compare their study's results with those of the ACKR3 study to highlight the current study's significance.

9) Page 9, line 257 "The binding mode of PS372424 is similar to the intrinsic agonist CXCL11, while VUF11222 inserts deeper and triggers a distinct conformational change in Trp2686.48. CXCL11 and PS372424 were reported to be agonists biased toward receptor internalization³¹. The similar conformation of Trp2686.48 in CXCR3 activated by CXCL11 and PS372424 may provide a structural explanation."

The conformation of side chain Trp268 seems to be different in the VUF11222-bound structure to accommodate the biaryl moiety. However, it is not clear how this observation provides an explanation for the biased agonist activity of CXCL11 and PS372424.

Minor points:

1) Page 4, line 85 "To facilitate the determination of the inhibited CXCR3 structure through cryo-EM ..." It is suggested to use antagonist bound CXCR3 instead of inhibited CXCR3.

2) Page 5, line 123-124 "suggesting that the salt bridge interactions between the KRGR motif and the receptor are crucial for receptor activation." should be revised to "suggesting that the salt bridge interactions between the KRGR motif and the receptor are crucial for CXCL11-induced receptor activation" to make it clear that it pertains specifically to CXCL11.

3) Page 5, line 129 "diminished CXCL11 activity on CXCR3," It might be more appropriate to use the term "CXCL11 affinity" instead of "activity" in this context.

4) I recommend including a WT curve in figure 2F, as in fig 2C, to allow for a better comparison.

5) Page 5, line 134 "PS372424 is a peptidomimetic agonist of CXCR3, consisting of a cyclohexanemethylamine group, a natural amino acid Arg, an unnatural amino acid Tic, and a deamino-Asp(Ph) group (Fig. 3a)." Need to add a citation for this compound.

6) Page 6, line 166 "In contrast to the deeply buried biaryl group, the bicycloaliphatic group only forms weak hydrophobic interactions with the sidechains of Trp1092.60 and Tyr601.39 (Fig. 4c)." It would be helpful to include the EC50 values for the W109A and Y60A mutants, described in Table S2, in the main text to support this statement.

Additionally, it would be useful to add the corresponding curves of these mutants to the figure 4d where this interaction is depicted.

7) Page 7, line 175 "The binding patterns of CXCL11 and PS372424 are also very similar, with all TMs of CXCR3 could be aligned well, and only a subtle conformational difference in the ECL2 loop could be observed (Fig. 5b). The TMs of VUF11222-activated CXCR3 go through a more distinct conformational change than CXCL11 or PS372424-activated CXCR3." Please provide RMSD values to support this statement as it would help to better understand the extent of conformational differences observed between CXCL11/PS372424-activated CXCR3 and VUF11222-activated CXCR3.

8) Page 9, line 254 "As no structure of chemokine receptors activated by small molecule agonists is available at present, the structures of CXCR3 coupled with small molecule agonists PS372424 and VUF11222 provide an opportunity for comparing the binding pattern of small molecule agonists with chemokines."

Typo "binding pattern" should be binding pattern.

9) Extended data Table 1, the % favored and outliers in Ramachandran plot appears to be listed in the opposite manner.

10) Extended data Table 2, please include the unit for pEC50. Please also make the use of significant figures realistic and consistent in all columns of the table (vary from 2-4 in pEC50 column and 1-3 in Fold column). For example, 2 for the pEC50 value and 1 for SEM (which probably should be reported as SD instead) is probably more accurate way of listing how reproducible the data is.

11) Include in the table a column that lists the specific number of measurements used for each data point. Also, the table legend says 2-4, but 2 measurements is probably not sufficient for attaining accurate values. Hopefully this only pertains to a few of the data points.

Reviewer #2:

Remarks to the Author:

This paper contains important information regarding the structure of chemokine receptor CXCR3/G protein in complex with chemokine CXCL11, small molecule agonists PS72424 and VUF1122; and a CXCR3 chimeric receptor in complex with antagonist SCH546738. These complexes reveal some unique features that are of interest to the field and are clearly discussed in the manuscript. However, I have concerns about the study in its present form, which I address below. However, even if my concerns are addressed, I still wonder about the suitability of this manuscript for NSMB.

1. Y60A and W109A receptor variants express to the same degree as WT CXCR3 on the cell surface, however, all other variants, when considering the large SD have fewer than 40% or less receptors on the cell surface when compared to WT (extended data Fig. 4). For example, Y271A is barely expressed on the cell surface (extended data Fig. 4) and yet it is able to robustly inhibit cAMP production, albeit with reduced potency and efficacy, relative to WT (Fig. 4D). This makes me wonder about the validation of the key interaction sites in terms of whether the altered signaling is due to binding defects. Why not directly

test binding using a pharmacological approach?

2. A chimeric CXCR3-KOR was generated by replacing the ICL3 of CXCR3 with ICL3 from KOR to facilitate structural determination because nanobody Nb6, which recognizes the ICL3 of κ OR, could be used to stabilize a complex with antagonist SCH546738 (Extended Data Fig. 1j-l). This was important to discover the extra density between TM5 and TM6 that could accommodate antagonist SCH546738. Although this strategy is clever, one wonders about the realistic nature of this presumed allosteric site, given the extensive engineering required to solve this structure.

3. The rationale for suggesting an allosteric site for SCH546738 is interesting, although not directly observed. Although potentially exciting, this lack of direct evidence diminishes enthusiasm for this possibility. Also, key validation/orthogonal experiments have not been pursued.

4. An important feature of the N-terminus of CXCR3 is that the proximal 16 amino acid residues are critical for recognition by CXCL11. A major limitation of the structure with CXCL11 is that the densities of the N-terminus (residues 1-39) of CXCR3 were not observed, thereby missing key structural insight that can further explain receptor-ligand interactions.

5. I'm curious about the other CXCR3 chemokines CXCL9 and CXCL10. Do they show a similar interaction pose as CXCL11? Can this be tested experimentally and/or computationally? How is their binding/signaling impacted by the various mutations that have been introduced here in CXCR3? It seems as though there is something to learn from such experiments/discussion.

Author Rebuttal to Initial comments

**Reviewer #1:**

**Remarks to the Author:**

**This study provides a comprehensive analysis of four CXCR3 structures bound to various**
**ligands, including an endogenous chemokine, two small molecule agonists, and an**
**antagonist. The authors confirmed the binding interfaces of each agonist on CXCR3**
**using mutagenesis and discovered an allosteric binding site in the antagonist-bound**
**structure that had not been previously identified in class A GPCRs. While these findings**
**offer valuable insights into receptor-ligand interaction, a more in-depth exploration of**
**the mechanisms beyond the binding interface and the newly discovered allosteric binding**
**sites would further enhance the study's impact. To achieve this, it is recommended that**
**the manuscript discusses the impact of these different binding modes on receptor**
**activation and signaling, as well as their potential applications in future drug**
**development efforts.**

**Response to the reviewer:**

We appreciate very much for your constructive comments and suggestions on our manuscript
“Structural insights into the activation and inhibition of CXC chemokine receptor 3” (ID:
NSMB-A47307-T). During the revision, the resolution of the CXCR3^{KOR}-SCH546738-Nb6
complex has been improved to 3.6 Å by combining newly collected CryoEM data. The density
map and the coordinate have been updated in the PDB data bank. The updated map and
coordinate will not influence our conclusions and could be provided if requested by the
reviewer. Point-to-point responses are provided below for your further consideration. We have
tried our best to revise our manuscript according to the comments, and all changes made to the
paper are highlighted in yellow color in the revised manuscript.

**Major points:**

**1) Page 2, line 22 “Structural analysis reveals that PS372424 shares a similar orthosteric**
**binding pocket with the N terminus of CXCL11, while VUF11222 buries deeper and**
**activates the receptor in a different way.”**

**The authors concluded that CXCL11 and PS372424 have a similar binding pocket, while**
**VUF11222 binds deeply, resulting in a distinct activation mechanism for the receptor.**
**While the structures do show conformational changes in the extracellular domain to**

accommodate different ligands, the intracellular sites appear to be highly similar across
all three structures. I would appreciate further clarification on the statement that
VUF11222 activates the receptor in a distinct manner. I am also interested in learning
more about the downstream signaling regulation properties of the three agonists
discussed in the paper. I am curious about whether they are neutral or biased agonists,
whether they have similar affinity to the receptor and whether the structures provide a
possible explanation for their unique properties.

**Response to the reviewer:**

Thank you very much for the constructive comments. The corresponding description
“VUF11222 buries deeper and activates the receptor in a different way” (page 2, line 22) has
been corrected to “VUF11222 buries deeper and activates the receptor in a distinct manner” in
the revised manuscript (page 2, line 23).

According to previous studies, CXCL11 is more efficacious at activating G_i signaling and β -
arrestin signaling than CXCL10 and CXCL9, and is slightly biased toward β -arrestin
signaling^{1,2}. PS372424 is efficacious at inducing receptor internalization, while its analog
VUF10661 (Response Figure 1.1A) was reported to recruit β -arrestin with greater efficacy than
CXCL11^{3,4}. The biased signaling of VUF11222 has not been studied, but its analog VUF11418
(Response Figure 1.1B) is reported to be a G-protein biased agonist^{5,6}. We also validated the
G_i dissociation and β -arrestin recruitment kinetics of CXCR3 in the presence of CXCL11,
PS372424, and VUF11222 (Response Figure 1.2A, B). As the bias signaling of CXCL11 has
been validated, the data of CXCL11 is used as a contrast. In both assays, the potency of
PS372424 is similar to that of CXCL11, while the efficacy of PS372424 increases (with a
greater increase in the efficacy for β -arrestin recruitment) (Response Figure 1.2A, B).
Therefore, similar to CXCL11, PS372424 appears to be biased towards β -arrestin signaling.
For VUF11222, increases in both $E_{max}^{G_i}$ and $EC_{50}^{G_i}$ are observed (Response Figure 1.2A, B),
thus it is difficult to estimate the signaling bias of VUF11222. Systematic works in the future
are required to illustrate the bias signaling of these three ligands. A proposed structural
explanation for the bias signaling was presented in the response to major point 9.

 **Response Figure 1.1 Chemical structures of PS372424, VUF10661, VUF11222,**
 **and VUF11418. (A) Chemical structures of PS372424 and VUF10661. (B)**
 **Chemical structures of VUF11222 and VUF11418. The differences in the structures**
 **are indicated by red circles.**

 **Response Figure 1.2 Bias signaling of CXCL11, PS372424, and VUF11222.**

The affinity of CXCL11 for CXCR3 measured by radiolabeled CXCL11 displacement binding
 assay is 0.3 nM^7 . The affinity of PS372424 measured by radiolabeled CXCL10 displacement
 assay is $42 \pm 21 \text{ nM}^8$. The pK_i value of VUF11222 measured by radiolabeled CXCL10
 displacement binding assay is 7.2 ± 0.1 (about 63 nM in K_i value)⁹. Analysis by PISA shows
 that the buried surface areas for CXCL11, PS372424, and VUF11222 are 1473.7 \AA^2 , 521.0 \AA^2 ,
 and 437.2 \AA^2 , respectively. A larger buried surface area generally indicates higher binding

affinity. Therefore, the reported binding affinities of CXCL11, PS372424, and VUF11222
match well with our structure analysis. A paragraph has been added in the revised manuscript
to illustrate the binding affinity of CXCL11, PS372424, and VUF11222 (page 7, line 189-192)

**2) In order to obtain the antagonist-bound structure, the authors generated a chimeric**
**protein that involved swapping the ICL3 of CXCR3 with the ICL3 from the kappa-opioid**
**receptor. However, the paper lacks any biochemical data that functionally characterizes**
**this chimeric protein. Furthermore, there is no mutagenesis data to verify the observe**
**allosteric binding site for the antagonist in the wild-type protein, as was done for the**
**agonists in cell-based assays. These omissions should be addressed experimentally.**

**Response to the Reviewer:**

Thank you for the constructive comments. The function of chimeric CXCR3^{KOR} has been
evaluated by cAMP assay. As shown in Response Figure 1.3, the EC₅₀ value is 1.60 nM for
CXCR3^{KOR} and 2.45 nM for wild-type CXCR3. Therefore, the activity of chimeric CXCR3^{KOR}
is comparable to that of the wild-type CXCR3. A sentence has been added in the revised
manuscript (page 4, line 89-91) and a panel has been added in Extended Figure 4 to illustrate
the activity of the chimeric protein.

The allosteric binding site has been evaluated by mutagenesis in the revised manuscript.
Residues Val261 and Ala265 were mutated to phenylalanine, which contains a larger
hydrophobic side chain. The mutations were expected to reduce the space of the allosteric
binding site and interfere with the binding of SCH546738. The CXCR3^{V261F} and CXCR3^{A265F}
mutants retain receptor activity as shown in Response Figure 1.4A-D. The EC₅₀ for
CXCR3^{V261F} is 3.33 nM, a value comparable with that of CXCR3^{WT} (Response Figure 1.4D).
The EC₅₀ for CXCR3^{A265F} increases to 34.2 nM (Response Figure 1.4D). For CXCR3^{WT}, an
increase in the SCH546738 concentration resulted in a decrease in the potency of CXCL11
(Response Figure 1.4A). For CXCR3^{V261F} and CXCR3^{A265F}, in contrast, the antagonism of
SCH546738 is less evident (Response Figure 1.4B, C). The assay indicates that mutation of
Val261 and Ala265 to phenylalanine may interfere with the binding of SCH546738.
Corresponding results have been added in the revised manuscript (page 9, line 244-252), and
three panels have been added in Figure 6e-g.

Response Figure 1.3 Validation of the activity of CXCR3^{KOR}. cAMP responses of CXCR3 and CXCR3^{KOR} to CXCL11. cAMP responses are normalized to the percent agonist activity of wild-type CXCR3.

Response Figure 1.4 Validation of the allosteric binding site. (A) cAMP responses of CXCR3 to CXCL11 in the presence of SCH546738 at different concentrations. (B) cAMP responses of CXCR3^{V261F} to CXCL11 in the presence of SCH546738 at different concentrations. (C) cAMP responses of CXCR3^{A265F} to CXCL11 in the presence of SCH546738 at different concentrations. (D) EC₅₀ values summarized for CXCR3, CXCR3^{V261F}, and CXCR3^{A265F}.

**3) Page 4, line 99 “The interactions between the proximal N-terminus (chemokine**
**recognition site 1, CRS1) of the receptor and the core domain (chemokine site 1, CS1) of**
**chemokine were reported to be weak and reversible.”**

**Further clarification is needed on what is meant by “reversible” in this context. Please**
**provide a citation to the original paper that supports this claim, rather than relying on a**
**review paper.**

**Response to the reviewer:**

Thank you for the constructive comments. The interactions between the proximal N-terminus
of the chemokine receptor (known as chemokine recognize site 1.0, CRS1.0) and the core
domain of the chemokine (known as chemokine site 1.0, CS1.0) have been widely studied in
the chemokine receptor family. In the case of CXCR2, although the interaction between
CRS1.0 and CS1.0 has been demonstrated, the N-terminus of CXCR2 was not traced in the
CryoEM structure of CXCR2 complexed with CXCL8¹⁰. To investigate the interaction
between the proximal N-terminus of CXCR3 and CXCL11, coarse-grained (CG) molecular
dynamics simulations were performed. CXCL11 recruitment could be observed in half of 20
independent CG simulations, and one of them is presented in Extended Data Fig 5 (also
presented in Response Figure 1.5). Therefore, the N-terminus of CXCR3 may play a key role
in the initial recruitment of CXCL11. The corresponding description has been included in the
revised manuscript (page 4, line 101-109).

A paragraph was included in the Method section (page 32, line 731-742) to illustrate the
simulation process as follow “The missing residues 1-39 of CXCR3 were modeled using
AlphaFold structure, while the CXCL11 was placed at least 30 Å away from CXCR3 structure.
The system was mapped into the EIneDyn22 CG model using the CHARMM-GUI Martini
Maker¹¹. The system was embedded in a 150 Å x 150 Å membrane, which is composed of 90%
1,2-dioleoyl-sn-glycero-3-phosphocholine and 10% cholesterol lipids. The system was then
further solvated in salt water of 150 mM salt concentration. Prior to the production run, a 5000-
step energy minimization and a restrained 5-ns NPT equilibration were performed. Then, 1 μs
production run was performed using the velocity rescaling thermostat and the Parrinello-
Rahman barostat to maintain the temperature at 300 K and 1 bar. The integration time step is
chosen to be 20 fs. To avoid the N-terminus aggregation at the entrance of the binding pocket,
the backbone of residue 35-44 was restrained during the simulations¹⁰. 20 independent CG
simulations were conducted.”

**Response Figure 1.5 The dynamic process of CXCL11 recruitment revealed by**
 **coarse-grained molecular dynamic simulation. The backbone of CXCR3 is shown**
 **as surface and colored violet, CXCL11 is shown as surface and colored gray.**

**4) Page 5, line 115 “Mutation of Trp1092.60, Phe1313.32, Phe1353.36, Tyr2716.51,**
 **Ser3047.39, and Tyr3087.43 results in reduced CXCL11-induced signaling response (Fig.**
 **2c, Extended Data Fig. 4, Extended Data Table 2), confirming the essential roles of these**
 **residues in CXCL11 binding and receptor activation.”**

**The data presented in Figure 2c suggests that the mutants exhibit the same maximum**
 **signal as the wild-type protein, indicating that there is no change in CXCL11-induced**
 **signaling responses, but potentially reduced binding affinity. The mutations made on the**
 **receptor are helpful in validating the binding interface between the receptor and**
 **CXCL11. However, to gain a more complete understanding of the observed interaction,**
 **it would be useful to investigate the role of the residues on the chemokine. Therefore, it**
 **would be worth summarizing existing literature on how mutations in CXCL11 affect**
 **CXCR3 signaling and compare them to the structural observations.**

**Response to the Reviewer:**

Thank you for the insightful suggestions. The importance of the N-terminus of CXCL11 has
 been extensively studied through mutation. Truncation of dipeptide in the N-terminus of
 CXCL11 by CD26 resulted in reduced CXCR3-binding property and chemotactic potency^{12,13}.
 A potent antagonist for CXCR3 could be obtained by truncation of three or four amino acids
 in the N-terminus of CXCL11, which lose the agonistic activity but retain marked binding

affinity^{14,15}. Mutation of ⁵KRGR⁸ to ⁵AAGA⁸, ⁴⁶KENKGQR⁵² to ⁴⁶AENAGQA⁵²,
⁵⁷KSKQAR⁶² to ⁵⁷ASAQAA⁶², and ⁶⁶KKVERK⁷¹ to ⁶⁶AAVEAA⁷¹ lead to 225, 30, 3, and 6-
184 fold loss in binding affinity compared to wild-type CXCL11¹⁶. These studies suggest that both
the ¹FPMF⁴ motif and the ⁵KRGR⁸ motif in the N-terminus of CXCL11 are essential for
receptor binding and activation. Several sentences have been added in the revised manuscript
(page 5, line 123-126, and line132-134).

**5) The numbering of residues in CXCL11 and receptor in the structure of SCH546738**
**presented in the provided PDB files do not match the numbering described in the**
**manuscript. For instance, residue E40 in the manuscript is labeled as E61 in the provided**
**PDB file. This discrepancy can be quite confusing and needs to be addressed by correcting**
**the numbering in the PDB file or vice versa.**

**Response to the Reviewer:**

Thank you for pointing this out. In the previous PDB file, the residues of CXCL11 were
numbered starting from the signal peptide. In the revised PDB file of CXCR3-CXCL11-DNGi-
scFv16, the residues of CXCL11 have been renumbered to be consistent with the figures.

For the numbering of CXCR3 in the structure of CXCR3^{κOR}-SCH546738-Nb6, the mismatch
is caused by the replacement of ICL3. Compared to the ICL3 of CXCR3, the ICL3 of κOR
contains 8 extra residues, so the residue number after ICL3 shifts by 8. The only way to
renumber the residues is to apply a different chain ID for the fragment from κOR, but this may
cause some confusion when depositing the PDB file. Therefore, the numbering of CXCR3 has
not been changed in the PDB file of CXCR3^{κOR}-SCH546738-Nb6.

**6) P6, line 145 “In a previous study, three-dimensional alignments of PS372424 to**
**pentapeptides of CXCL10 suggest that PS372424 may mimic the residues 35-39 or 19-23**
**of CXCL1036.” And the conclusion from this paragraph: “we suggest the binding pattern**
**of PS372424 is similar to CXCL11 rather than CXCL10.”**

**The paper cited in the manuscript clearly states that “We emphasize that there is no**
**direct evidence that any small molecule agonist binds to CXCR3 in the same manner as**
**any portion of IP-10. Rather we present this example to show how three-dimensional**
**alignment can be used in a nonstandard manner to generate reasonable testable**

**hypotheses” The paper primarily focuses on the geometry of PS372424 and suggests that**
**its 3D conformation might be similar to either 35-39 or 19-23 of CXCL10. In fact, the**
**structure of PS372424 in your model aligns well with one of their predictions. Therefore,**
**it may not be appropriate to conclude that "the binding pattern of PS372424 is similar to**
**CXCL11 rather than CXCL10" based solely on the cited paper.**

**Response to the Reviewer:**

Thank you for pointing it out. We are sorry for our misinterpretation. The corresponding
description “we suggest the binding pattern of PS372424 is similar to CXCL11 rather than
CXCL10” (page 6, line 151-152) has been corrected to “In conclusion, we suggest that the
binding pattern of PS372424 and the N-terminal pentapeptides of CXCL11 share similarity”
(page 6, line 164-166).

**7) Page 7, line 184 “Compared with the VUF11222 coupled receptor, a kink in TM1 could**
**be observed in the CXCL11 coupled receptor, making the N-terminus of TM1 bend**
**toward TM7 (Fig.5c). “**

**In the structure of VUF11222, there is disorder in part of TM1 and the N-terminus when**
**compared to the CXCL11 and PS372424 structures. It is unclear what may have caused**
**this disorder, particularly since the N-terminus is tethered to TM7 by a disulfide bond in**
**the other two structures.**

**Response to the Reviewer:**

Thanks for the insightful comments. Looking through the structures of chemokine receptors,
we found that a similar phenomenon could be observed in other chemokine receptors. In the
structure of chemokine-CCR1/CCR2/CCR5-G_i complex, the disulfide bond between the N-
terminus of TM1 and TM7 could be traced (Response Figure 1.6A-C). However, in the apo-
CCR1/CCR3/CCR5-G_i structure, the disulfide bond could not be modeled (Response Figure
1.6A-C). This suggests that the disulfide bond alone may not be sufficient for stabilizing the
N-terminus of TM1. The interactions with the agonist may play an essential role.

In the unsharpened map of VUF11222-CXCR3-DNG_i (Response Figure 1.6D), weak and
discontinuous densities could be observed between TM1 and TM7, suggesting the flexibility
of the N-terminus of TM1. The presence of the disulfide bond is not sufficient to stabilize the

N-terminus of TM1. Additional interactions between the N-terminus of TM1 and
 CXCL11/PS372424 would help to further stabilize the conformation.

 **Response Figure 1.6 The disulfide bond between the N-terminus of TM1 and**
 **TM7. (A)** The disulfide bond is observed in the CCL15-CCR1-G_i complex but not
 the apo-CCR1-G_i complex. **(B)** The disulfide bond is observed in the CCL2-CCR2-
 G_i complex but not the apo-CCR3-G_i complex. **(C)** The disulfide bond is observed
 in the CCL3-CCR5-G_i complex but not the apo-CCR5-G_i complex. In B-D, the
 disulfide bonds are shown as spheres and indicated by red circles. **(D)** Weak densities
 between TM1 and TM7 indicate the bending of TM1 toward TM7.

**8) Page 9, line 254 “As no structure of chemokine receptors activated by small molecule**
 **agonists is available at present, the structures of CXCR3 coupled with small molecule**
 **agonists PS372424 and VUF11222 provide an opportunity for comparing the binding**
 **pattern of small molecule agonists with chemokines.”**

This statement is incorrect, and it would be appropriate to refer to a study that
demonstrates the structure of an atypical chemokine receptor (ACKR3) bound to a small
molecule agonist (PMID: 35857509). The author should cite this paper and make essential
revisions to their comment. It may also be advantageous for the author to compare their
study's results with those of the ACKR3 study to highlight the current study's significance.

**Response to the Reviewer:**

Thank you for pointing it out. The structure of ACKR3 complexed with a small molecule
CCX662 has been cited in the revised manuscript and the corresponding description has been
corrected to “At present, only one structure of chemokine receptor activated by small molecular
agonist was available, which is the structure of atypical chemokine receptor ACKR3
complexed with a small molecule agonist CCX662⁴⁹. The structures of CXCR3 coupled with
small molecule agonists PS372424 and VUF11222 would provide more information for
comparing the binding pattern of small molecule agonists with chemokines targeting the typical
chemokine receptors.” (page 10, line 289-291).

In addition, the structure of CXCL12 bound ACKR3 was included in Extended Data Fig. 6a
(Response Figure 1.7). A corresponding description has been added in the revised manuscript
as follow: “In addition, CXCL11 shares a binding pose similar to CXCL8 and CCL20 with
minor deviations. Compared to CXCL11, CCL5 and CCL3 undergo a rotation along a pseudo
axis parallel to the membrane surface, while CXCL12 undergoes a rotation along a pseudo axis
perpendicular to the membrane surface. The difference in the insertion depth and binding pose
suggests that the binding patterns of chemokines differ widely” (page 10, line 271-275).

**Response Figure 1.7 Comparison of the insertion depth of CXCL11 (violet),**
**CXCL8 (green), CCL20 (cyan), CCL3 (orange), CCL5 (yellow), and CXCL12**

(blue). The receptors are shown as cartoon and colored gray, while the chemokines
are shown in surface model. The pseudo rotation axis and orientation for CCL3,
CCL5, and CXCL12 compared to CXCL11 are shown.

**9) Page 9, line 257 “The binding mode of PS372424 is similar to the intrinsic agonist**
**CXCL11, while VUF11222 insets deeper and triggers a distinct conformational change in**
**Trp2686.48. CXCL11 and PS372424 were reported to be agonists biased toward receptor**
**internalization³¹. The similar conformation of Trp2686.48 in CXCR3 activated by**
**CXCL11 and PS372424 may provide a structural explanation.”**

**The conformation of side chain Trp268 seems to be different in the VUF11222-bound**
**structure to accommodate the biaryl moiety. However, it is not clear how this observation**
**provides an explanation for the biased agonist activity of CXCL11 and PS372424.**

**Response to the Reviewer:**

Thank you very much for the constructive comments. In the absence of a structure of β -arrestin
coupled CXCR3, it is difficult to interpret the molecular basis of bias signaling of CXCR3.
Structural analysis of available structures of β -arrestin coupled GPCRs may give some clues.
When superposed with the structure of β -arrestin coupled 5HT2B, a micro-tilt in the lower half
of TM7 (named TM^{7.47-7.53}) is observed in the structure of G_q-coupled 5HT2B (Response
Figure 1.8A)¹⁷. Superposition of β -arrestin-coupled with G_o-coupled ACM2 and superposition
of β -arrestin-coupled with G_q-coupled NTR1 show similar micro-tilt in TM^{7.47-7.53} (Response
Figure 1.8B, C)¹⁸⁻²¹. Therefore, it appears that TM^{7.47-7.53} tilt to a greater degree in the G
protein-coupled receptor than in the β -arrestin-coupled receptor. The micro-tilt in TM^{7.47-7.53}
extends the side chain of 7.53 (a conserved Tyr in the NPxxY motif) into a different spatial
position in the intracellular binding cavity. Since residue 7.53 is sandwiched between TM3 and
TM6, the spatial position of 7.53 may affect the overall property of the signaling protein
binding cavity and ultimately influence the signaling bias.

In our structures, no obvious difference could be observed in the spatial position of 7.53. This
is mainly due to the coupling of G protein in all three structures (Response Figure 1.8D).
However, a micro-tilt at 7.47 is observed in the VUF11222-coupled CXCR3 when superposed
with the CXCL11- and PS372424-coupled CXCR3 (Response Figure 1.8D). As shown in
Response Figure 1.8E, the insertion of VUF11222 causes a microswitch in Trp268 that repels
His310 away. Together with the repulsion of Ser307 and Tyr308 directly caused by VUF11222,

the N-terminus of TM7^{7.47-7.53} is tilted (Response Figure 1.8E). A possible mechanism for the
 biased signaling of CXCR3 is shown in Response Figure 1.8F. The insertion of VUF11222
 initiates the tilt in the N-terminus of TM7^{7.47-7.53}, which is transferred to the C-terminus of
 TM7^{7.47-7.53}, moves Tyr318^{7.53} towards the central of the signaling protein binding cavity, and
 ultimately influence the binding of signaling proteins.

A paragraph has been included in the Discussion section to discuss the mechanism for the
 biased signaling of CXCR3 (page 11, line 297-327). We are struggling to determine the
 structure of CXCR3 coupled with β -arrestin and expect to validate the hypothesis in the future.
 In addition, discovering of biased agonists will also be helpful for understanding the bias
 agonism of CXCR3.

 **Response Figure 1.8 A proposed mechanism for the bias signaling of CXCR3.**
 (A) Superposition of the β -arrestin-coupled and G protein-coupled 5HT2B. (B)
 Superposition of the β -arrestin-coupled and G protein-coupled ACM2. (C)
 Superposition of the β -arrestin-coupled and G protein-coupled NTR1. In A-C, the
 receptors are shown as cartoon, and colored cyan for β -arrestin-coupled receptor and
 gray for G protein-coupled receptor. The CA atoms of 7.47 and 7.53 are shown as
 spheres, and the side chains of 7.53 are shown as sticks. The tilt in TM7^{7.47-7.53} is

indicated by black arrows. **(D)** Superposition of G_i-coupled CXCR3 activated by
CXCL11 (violet), PS372424 (yellow), and VUF11222 (blue). **(E)** The tilt of TM7^{7.47-}
345 ^{7.53} initiated by the insertion of VUF11222. **(F)** A proposed model for the mechanism
for the biased signaling of CXCR3.

**Minor points:**

**1) Page 4, line 85 “To facilitate the determination of the inhibited CXCR3 structure**
**through cryo-EM ...” It is suggested to use antagonist bound CXCR3 instead of inhibited**
**CXCR3.**

**Response to the Reviewer:**

Thank you very much for the kind suggestion. The description “To facilitate the determination
of the inhibited CXCR3 structure through cryo-EM single particle analysis” (page 4, line 85)
in the previous manuscript has been revised to “To facilitate the determination of the antagonist
bound CXCR3 structure through cryo-EM single particle analysis” (page 4, line 85).

**2) Page 5, line123-124 “suggesting that the salt bridge interactions between the KRGR**
**motif and the receptor are crucial for receptor activation.” should be revised to**
**“suggesting that the salt bridge interactions between the KRGR motif and the receptor**
**are crucial for CXCL11-induced receptor activation” to make it clear that it pertains**
**specifically to CXCL11.**

**Response to the Reviewer:**

Thank you very much for the kind suggestion. The description “suggesting that the salt bridge
interactions between the KRGR motif and the receptor are crucial for receptor activation” (page
5, line 123-124) in the previous manuscript has been revised to “suggesting that the salt bridge
interactions between the KRGR motif and the receptor are crucial for CXCL11-induced
receptor activation” (page 5, line 136-137).

3) Page 5, line 129 “diminished CXCL11 activity on CXCR3,” It might be more
appropriate to use the term “CXCL11 affinity” instead of “activity” in this context.

**Response to the Reviewer:**

Thank you very much for the kind suggestion. The description “Mutation of Arg197^{ECL2} and
Arg212^{5.35} diminished CXCL11 activity on CXCR3” (page5, line129) in the previous
manuscript has been revised to “Mutation of Arg197^{ECL2} and Arg212^{5.35} results in reduced
CXCL11 binding affinity” (page 6, line 142).

4) I recommend including a WT curve in figure 2F, as in fig 2C, to allow for a better
comparison.

**Response to the Reviewer:**

Thank you very much for the kind suggestion. A WT curve has been included in Figure 2F
(Also presented in Response Figure 1.9).

**Response Figure 1.9 cAMP responses of CXCR3 mutants to CXCL11.** cAMP
responses are normalized to the percent agonist activity of wild-type CXCR3.

5) Page 5, line 134 “PS372424 is a peptidomimetic agonist of CXCR3, consisting of a
cyclohexanemethylamine group, a natural amino acid Arg, an unnatural amino acid Tic,
and a deamino-Asp(Ph) group (Fig. 3a).” Need to add a citation for this compound.

**Response to the Reviewer:**

Thank you very much for the kind suggestion. The paper which identified the agonist
PS372424 (“Stroke, I. L. *et al.* Identification of CXCR3 receptor agonists in combinatorial
small-molecule libraries. *Biochem Biophys Res Commun* **349**, 221-228,
doi:10.1016/j.bbrc.2006.08.019 (2006).”) has been cited (page 6, line 148).

**6) Page 6, line 166 “In contrast to the deeply buried biaryl group, the bicycloaliphatic**
**group only forms weak hydrophobic interactions with the sidechains of Trp1092.60 and**
**Tyr601.39 (Fig. 4c).” It would be helpful to include the EC₅₀ values for the W109A and**
**Y60A mutants, described in Table S2, in the main text to support this statement.**
**Additionally, it would be useful to add the corresponding curves of these mutants to the**
**figure 4d where this interaction is depicted.**

**Response to the Reviewer:**

Thank you very much for the kind suggestion. A sentence “As comparison, mutation of
Trp109^{2.60} and Tyr60^{1.39} results in no obvious changes in the EC₅₀ values (Fig. 4d, Extended
Data Fig. 4, Extended Data Table 2).” has been added in the revised manuscript (page 7, line
184-186). The corresponding curves of Trp109^{2.60} and Tyr60^{1.39} mutant have been included in
Figure 4d as well (Also presented in Response Figure 1.10).

**Response Figure 1.10 cAMP responses of CXCR3 mutants to VUF11222. cAMP**
**responses are normalized to the percent agonist activity of wild-type CXCR3.**

**7) Page 7, line 175 “The binding patterns of CXCL11 and PS372424 are also very similar,**
**with all TMs of CXCR3 could be aligned well, and only a subtle conformational difference**
**in the ECL2 loop could be observed (Fig. 5b). The TMs of VUF11222-activated CXCR3**
**go through a more distinct conformational change than CXCL11 or PS372424-activated**
**CXCR3.” Please provide RMSD values to support this statement as it would help to better**
**understand the extent of conformational differences observed between**
**CXCL11/PS372424-activated CXCR3 and VUF11222-activated CXCR3.**

**Response to the Reviewer:**

Thank you very much for the kind suggestion. The RMSD between PS372424- and CXCL11-
activated CXCR3 is 0.398 Å, while the RMSD between VUF11222- and CXCL11-activated
CXCR3 is 0.938 Å, and the RMSD between VUF11222- and PS372424-activated CXCR3 is
0.984 Å. The RMSD values further confirm that VUF11222-activated CXCR3 adopts a
distinct conformation compared to CXCL11- and PS372424-activated CXCR3. A sentence
“The RMSD (Root-Means-Square Deviation) between PS372424- and CXCL11-activated
CXCR3 is 0.398 Å, while the RMSD between VUF11222- and CXCL11/PS372424-activated
CXCR3 is 0.938/0.984 Å.” has been added in the revised manuscript (page 7, line 197-200).

**8) Page 9, line 254 “As no structure of chemokine receptors activated by small molecule**
**agonists is available at present, the structures of CXCR3 coupled with small molecule**
**agonists PS372424 and VUF11222 provide an opportunity for comparing the binging**
**pattern of small molecule agonists with chemokines.”**

**Typo “binging pattern” should be binding pattern.**

**Response to the Reviewer:**

Thank you for pointing it out. We are very sorry for our incorrect writing. The typo “binging
pattern” (page 9, line256) has been corrected as “binding pattern” (page 10, line 292).

**9) Extended data Table 1, the % favored and outliers in Ramachandran plot appears to**
**be listed in the opposite manner.**

**Response to the Reviewer:**

Thank you for pointing it out. We are very sorry for our incorrect writing. The error in the %
favored and outliers in the Ramachandran plot in Extended Data Table 1 has been corrected.

**10) Extended data Table 2, please include the unit for pEC50. Please also make the use of**
**significant figures realistic and consistent in all columns of the table (vary from 2-4 in**
**pEC50 column and 1-3 in Fold column). For example, 2 for the pEC50 value and 1 for**
**SEM (which probably should be reported as SD instead) is probably more accurate way**
**of listing how reproducible the data is.**

**Response to the Reviewer:**

Thank you for your correction. The Extended Data Table has been updated, and the $\log EC_{50} \pm$
Std.error and EC_{50} (M) values calculated in Graphpad Prism are listed. 3 significant figures
were used for the $\log EC_{50}$ and EC_{50} values. 2 significant figures were used for the Std.error of
$\log EC_{50}$ and the fold values. As explained by GraphPad Prism
([https://www.graphpad.com/support/faq/why-does-prism-report-the-standard-error-for-
logec50-but-not-for-the-ec50-itself-when-it-fits-dose-response-curves/](https://www.graphpad.com/support/faq/why-does-prism-report-the-standard-error-for-logec50-but-not-for-the-ec50-itself-when-it-fits-dose-response-curves/)), the Std.error of EC_{50}
could not be calculated and thus not listed in the revised Table.

**11) Include in the table a column that lists the specific number of measurements used for**
**each data point. Also, the table legend says 2-4, but 2 measurements is probably not**
**sufficient for attaining accurate values. Hopefully this only pertains to a few of the data**
**points.**

**Response to the Reviewer:**

Thank you very for the kind suggestion. The values of measurements have been included in
Extended Data Table 2 in the revised manuscript. It should be noted that the measurements for
CXCL11 have been replenished to three measurements. For the revised EC_{50} values for
CXCL11, small variations are observed but have little impact on our conclusion.

**Reviewer #2:**

**Remarks to the Author:**

**This paper contains important information regarding the structure of chemokine**
**receptor CXCR3/G protein in complex with chemokine CXCL11, small molecule agonists**
**PS72424 and VUF1122; and a CXCR3 chimeric receptor in complex with antagonist**
**SCH546738. These complexes reveal some unique features that are of interest to the field**
**and are clearly discussed in the manuscript. However, I have concerns about the study in**
**its present form, which I address below. However, even if my concerns are addressed, I**
**still wonder about the suitability of this manuscript for NSMB.**

**Response to the reviewer:**

We appreciate very much for your constructive comments and suggestions on our manuscript
“Structural insights into the activation and inhibition of CXC chemokine receptor 3” (ID:
NSMB-A47307-T). During the revision, the resolution of the CXCR3^{KOR}-SCH546738-Nb6
complex has been improved to 3.6 Å by combining newly collected CryoEM data. The density
map and the coordinate have been updated in the PDB data bank. The updated map and
coordinate will not influence our conclusions and could be provided if requested by the
reviewer. Point-to-point responses are provided below for your further consideration. We have
tried our best to revise our manuscript according to the comments, and all changes made to the
paper are highlighted in yellow color in the revised manuscript.

**1. Y60A and W109A receptor variants express to the same degree as WT CXCR3 on the**
**cell surface, however, all other variants, when considering the large SD have fewer than**
**40% or less receptors on the cell surface when compared to WT (extended data Fig. 4).**
**For example, Y271A is barely expressed on the cell surface (extended data Fig. 4) and yet**
**it is able to robustly inhibit cAMP production, albeit with reduced potency and efficacy,**
**relative to WT (Fig. 4D). This makes me wonder about the validation of the key**
**interaction sites in terms of whether the altered signaling is due to binding defects. Why**
**not directly test binding using a pharmacological approach?**

**Response to the Reviewer:**

Thanks a lot for the constructive comments. The binding assay is very useful to illustrate the
activity of CXCR3 mutants, however, the assay was not performed in our research for two

reasons. Firstly, the CXCR3 apo-protein is unstable and couldn't be purified without the
 addition of antagonists or agonists. Therefore, it is less likely to conduct a binding assay in
 vitro using purified CXCR3 protein. And secondary, radiolabeled CXCL11, PS372424, and
 VUF11222 are not commercialized and thus are very difficult to obtain.
 Alternatively, we use a computational method to estimate the binding energy change upon
 different mutations. Here we use "molecular mechanics-Poisson Boltzmann surface area"
 calculations. In short, if the mutation only affects the ligand binding but does not contribute to
 conformational dynamics, the trend of EC50 increased fold over delta (delta H) should be linear.
 As the Response Figure 2.1 suggested, except for F135A, other residues likely affect ligand
 binding most. F135A locates in the middle of TM3 and bottom of the orthosteric pocket,
 closely associates with W268, which is an essential residue for receptor activation, and may
 contribute to both the ligand binding and conformational change of the receptor.

 **Response Figure 2.1 Molecular mechanics-Poisson Boltzmann surface area**
 **calculations of CXCR3 mutants bound to three agonists.** The trend of EC50
 increased fold over delta (delta H) is linear except for F135A in three cases.

**2. A chimeric CXCR3-KOR was generated by replacing the ICL3 of CXCR3 with ICL3**
 **from KOR to facilitate structural determination because nanobody Nb6, which**
 **recognizes the ICL3 of κOR, could be used to stabilize a complex with antagonist**
 **SCH546738 (Extended Data Fig. 1j-l). This was important to discover the extra density**
 **between TM5 and TM6 that could accommodate antagonist SCH546738. Although this**
 **strategy is clever, one wonders about the realistic nature of this presumed allosteric site,**
 **given the extensive engineering required to solve this structure.**

**Response to the Reviewer:**

Thank you for the constructive comments. The function of chimeric CXCR3^{KOR} has been
 evaluated by cAMP assay. As shown in Response Figure 2.2, the EC50 value is 1.60 nM for

CXCR3^{KOR} and 2.45 nM for wild-type CXCR3. Therefore, the activity of chimeric CXCR3^{KOR}
is comparable to that of the wild-type CXCR3. A sentence has been added in the revised
manuscript (page 4, line 89-91) and a panel has been added in Extended Figure 4 to illustrate
the activity of the chimeric protein.

**Response Figure 2.2 Validation of the activity of CXCR3^{KOR}.** cAMP responses of
CXCR3 and CXCR3^{KOR} to CXCL11. cAMP responses are normalized to the percent
agonist activity of wild-type CXCR3. Data are shown as means \pm *S.E.M.*

As shown in Response Figure 2.3A, the chimeric fragment has little contribution to the
formation of the allosteric binding site. By comparing the amino acid residues around the
allosteric binding site in CXCR2, CXCR3, and CXCR4, the side chains of residue Ala265^{6,45}
in CXCR3 are smaller than Leu261^{6,45} and Phe249^{6,45} in CXCR2 and CXCR4 (Response
Figure 2.3B). In addition, a microswitch is observed in the side chain of Leu224^{5,47} in CXCR3
compared to Leu228^{5,47} in CXCR2, resulting in a larger opening between the TM5 and TM6 in
CXCR3. Within the pocket, CXCR3 also contains a smaller residue Ala139^{3,40} (Ile134^{3,40} in
CXCR2 and Val124^{3,40} in CXCR4 for comparison), making the space large enough to
accommodate SCH546738. In conclusion, the formation of the allosteric binding site is mainly
attributed to the constitutional and conformational uniqueness of the residues surrounding the
allosteric binding site in CXCR3.

**Response Figure 2.3 The allosteric binding site in CXCR3.** (A) The allosteric
 binding pocket is mainly contributed by CXCR3. CXCR3^{KOR} is shown as surface,
 the fragment from CXCR3 and the fragment from κOR are color violet and orange,
 respectively. SCH546738 is shown as spheres and colored blue. (B) The formation
 of the allosteric binding site in CXCR3 is mainly attributed to the small size of side
 chains around the pocket. The structure of CXCR3, CXCR2, and CXCR4 are shown
 as cartoon and colored violet, gray, and orange, respectively. The side chains around
 the allosteric binding pocket are shown as sticks.

**3. The rationale for suggesting an allosteric site for SCH546738 is interesting, although**
 **not directly observed. Although potentially exciting, this lack of direct evidence**
 **diminishes enthusiasm for this possibility. Also, key validation/orthogonal experiments**
 **have not been pursued.**

**Response to the Reviewer:**

The allosteric binding site has been evaluated by mutagenesis in the revised manuscript.
 Residues Val261 and Ala265 were mutated to phenylalanine, which contains a larger
 hydrophobic side chain. The mutations were expected to reduce the space of the allosteric
 binding site and interfere with the binding of SCH546738. The CXCR3^{V261F} and CXCR3^{A265F}
 mutants retain receptor activity as shown in Response Figure 2.4A-D. The EC₅₀ for
 CXCR3^{V261F} is 3.33 nM, a value comparable with that of CXCR3^{WT} (Response Figure 2.4D).
 The EC₅₀ for CXCR3^{A265F} increases to 34.2 nM (Response Figure 2.4D). For CXCR3^{WT}, an
 increase in the SCH546738 concentration resulted in a decrease in the potency of CXCL11

(Response Figure 2.4A). For CXCR3^{V261F} and CXCR3^{A265F}, however, the antagonism of
 SCH546738 is less evident (Response Figure 2.4B, C). The assay indicates that mutation of
 Val261 and Ala265 to phenylalanine may interfere with the binding of SCH546738.
 Corresponding results have been added in the revised manuscript (page 9, line 244-252), and
 three panels have been added in Figure 6e-g.

 **Response Figure 2.4 Validation of the allosteric binding site. (A)** cAMP responses
 of CXCR3 to CXCL11 in the presence of SCH546738 at different concentrations.
 **(B)** cAMP responses of CXCR3^{V261F} to CXCL11 in the presence of SCH546738 at
 different concentrations. **(C)** cAMP responses of CXCR3^{A265F} to CXCL11 in the
 presence of SCH546738 at different concentrations. **(D)** EC₅₀ values summarized for
 CXCR3, CXCR3^{V261F}, and CXCR3^{A265F}. Data are shown as means ± *S.E.M.*.

 **4. An important feature of the N-terminus of CXCR3 is that the proximal 16 amino acid**
 **residues are critical for recognition by CXCL11. A major limitation of the structure with**
 **CXCL11 is that the densities of the N-terminus (residues 1-39) of CXCR3 were not**
 **observed, thereby missing key structural insight that can further explain receptor-ligand**
 **interactions.**

**Response to the Reviewer:**

Thank you for the constructive comments. The interactions between the proximal N-terminus
of the chemokine receptor (known as chemokine recognize site 1.0, CRS1.0) and the core
domain of the chemokine (known as chemokine site 1.0, CS1.0) have been widely studied in
the chemokine receptor family. In the case of CXCR2, although the interaction between
CRS1.0 and CS1.0 has been demonstrated, the N-terminus of CXCR2 was not traced in the
CryoEM structure of CXCR2 complexed with CXCL8¹⁰. To investigate the interaction
between the proximal N-terminus of CXCR3 and CXCL11, coarse-grained (CG) molecular
dynamics simulations were performed. CXCL11 recruitment could be observed in half of 20
independent CG simulations, and one of them is presented in Extended Data Fig 5 (also
presented in Response Figure 2.5). Therefore, the N-terminus of CXCR3 may play a key role
in the initial recruitment of CXCL11. The corresponding description has been included in the
revised manuscript (page 4, line 101-109).

A paragraph was included in the Method section (page 32, line 731-742) to illustrate the
simulation process as follow “The missing residues 1-39 of CXCR3 were modeled using
AlphaFold structure, while the CXCL11 was placed at least 30 Å away from CXCR3 structure.
The system was mapped into the EIneDyn22 CG model using the CHARMM-GUI Martini
Maker¹¹. The system was embedded in a 150 Å x 150 Å membrane, which is composed of 90%
1,2-dioleoyl-sn-glycero-3-phosphocholine and 10% cholesterol lipids. The system was then
further solvated in salt water of 150 mM salt concentration. Prior to the production run, a 5000-
step energy minimization and a restrained 5-ns NPT equilibration were performed. Then, 1 μs
production run was performed using the velocity rescaling thermostat and the Parrinello-
Rahman barostat to maintain the temperature at 300 K and 1 bar. The integration time step is
chosen to be 20 fs. To avoid the N-terminus aggregation at the entrance of the binding pocket,
the backbone of residue 35-44 was restrained during the simulations¹⁰. 20 independent CG
simulations were conducted.”

**Response Figure 2.5 The dynamic process of CXCL11 recruitment revealed by**
 **coarse-grained molecular dynamic simulation.** The backbone of CXCR3 is shown
 as surface and colored violet, CXCL11 is shown as surface and colored gray.

**5. I'm curious about the other CXCR3 chemokines CXCL9 and CXCL10. Do they show**
 **a similar interaction pose as CXCL11? Can this be tested experimentally and/or**
 **computationally? How is their binding/signaling impacted by the various mutations that**
 **have been introduced here in CXCR3? It seems as though there is something to learn**
 **from such experiments/discussion.**

**Response to the Reviewer:**

Thank you for the constructive comments. During the revision, several CXCR3 mutants have
 been tested for CXCL10. Among the mutants tested, mutations at D52, W109, F131, S304, and
 Y308 result in reduced potency of CXCL10 (Response Figure 2.6 and Response Table 2.1).
 However, the EC₅₀ fold increase for CXCL10 is generally lower than that for CXCL11. In
 addition, Y271A mutant has little effect on the potency of CXCL10, but reduces the potency
 of CXCL11. Therefore, we suggest that CXCL10 may bind to CXCR3 in a binding pocket that
 overlaps with that of CXCL11. However, the precise interaction network between CXCR3 and
 CXCL10 may be different from that between CXCR3 and CXCL11.

Response Figure 2.6 cAMP responses of CXCR3 mutants to CXCL10. cAMP responses are normalized to the percent agonist activity of wild-type CXCR3. Data are shown as means \pm *S.E.M.*.

666

667

Response Table 2.1 EC₅₀ values for CXCL11 and CXCL10.

EC ₅₀ (nM) for CXCL11				EC ₅₀ (nM) for CXCL10			
Mutants	EC ₅₀ (nM)	N	Fold	Mutants	EC ₅₀ (nM)	N	Fold
WT	2.25	4	1.0	WT	22.3	6	1.0
D52A	155	3	69	D52A	205	6	9.2
W109A	186	3	83	W109A	276	6	12
F131A	24.2	3	11	F131A	153	6	6.9
271A	11.9	3	5.3	Y271A	12.8	6	0.6
S304L	34.8	3	15	S304L	239	6	11
Y308A	9.46	3	4.2	Y308A	87.9	6	3.9

668

669

670 **Response References**

671

[revised manuscript text omitted]

733

734

Decision Letter, first revision:

Message: Our ref: NSMB-A47307A

12th Sep 2023

Dear Dr. Hu,

Thank you for submitting your revised manuscript "Structural insights into the activation and inhibition of CXC chemokine receptor 3" (NSMB-A47307A). It has now been seen by the original referees and their comments are below. The reviewers find that the paper has improved in revision, and therefore we'll be happy in principle to publish it in Nature Structural & Molecular Biology, pending minor revisions to satisfy the referees' final requests and to comply with our editorial and formatting guidelines.

To facilitate our work at this stage, it is important that we have a copy of the main text as a word file. If you could please send along a word version of this file as soon as possible, we would greatly appreciate it; please make sure to copy the NSMB account (cc'ed above).

Sincerely,

Katarzyna Ciazynska
(she/her)
Associate Editor
Nature Structural & Molecular Biology
<https://orcid.org/0000-0002-9899-2428>

Reviewer #1 (Remarks to the Author):

The majority of the comments I provided have been appropriately addressed by the author in the revised manuscript. I am satisfied with the modifications. There are a few minor suggestions that I would like to point out:

1) In fig. 4d, although Y60A exhibits a comparable EC50 to the WT, its efficacy (Emax) is significantly reduced. This trend is similarly observed in fig. 3d, where there's a reduction in affinity as well. Since the surface expression level of Y60A is comparable to that of WT, the reduced efficacy of Y60A might be worth discussing within the manuscript.

2) It would be beneficial to include the full name of PISA and provide a citation or the link

to its website. This would enable readers to easily access it.

3) Within the figure legend of Extended fig. 8, I recommend using "Ca atoms" instead of CA atoms.

Reviewer #2 (Remarks to the Author):

I am happy with the rebuttal and revisions. I have no additional comments.

Author Rebuttal, first revision:

Reviewer #1 (Remarks to the Author):

The majority of the comments I provided have been appropriately addressed by the author in the revised manuscript. I am satisfied with the modifications. There are a few minor suggestions that I would like to point out:

Response to the reviewer: We appreciate very much for your constructive comments and suggestions on our manuscript "Structural insights into the activation and inhibition of CXC chemokine receptor 3" (ID: NSMB-A47307A). Point-to-point responses are provided below for your further consideration, and changes made to the paper are highlighted in yellow color in the revised manuscript.

1) In fig. 4d, although Y60A exhibits a comparable EC50 to the WT, its efficacy (Emax) is significantly reduced. This trend is similarly observed in fig. 3d, where there's a reduction in affinity as well. Since the surface expression level of Y60A is comparable to that of WT, the reduced efficacy of Y60A might be worth discussing within the manuscript.

Response: Thank you very much for the constructive comments. A sentence “Notably, the mutation of Tyr60^{1.39} significantly reduced the E_{\max} of VUF11222, suggesting that the mutation may change the conformational landscape of the receptor or the binding kinetics of the ligand, perhaps by affecting the dissociation rate constant K_{off} .” has been added in the manuscript to discuss the reduced efficacy of Y60A. (page 7, line185-187)

2) It would be beneficial to include the full name of PISA and provide a citation or the link to its website. This would enable readers to easily access it.

Response: Thank you for pointing it out. The full name of PISA, which is “Proteins, Interfaces, Structures and Assemblies” as well as the citation “Krissinel, E. & Henrick, K. Inference of macromolecular assemblies from crystalline state. *J Mol Biol* 372, 774-797, doi:10.1016/j.jmb.2007.05.022 (2007).” has been added in the revised manuscript. (page 7, line 191)

3) Within the figure legend of Extended fig. 8, I recommend using “C α atoms” instead of CA atoms.

Response: The “CA atoms” has been corrected to “C α atoms” in the legend of Extended Fig. 8 in the revised manuscript.

Reviewer #2 (Remarks to the Author):

I am happy with the rebuttal and revisions. I have no additional comments.

Response to the reviewer: We appreciate very much for your constructive comments and suggestions on our manuscript "Structural insights into the activation and inhibition of CXC chemokine receptor 3" (ID: NSMB-A47307A).

Final Decision Letter:

Message 3rd Nov 2023

:

Dear Dr. Hu,

We are now happy to accept your revised paper "Structural insights into the activation and inhibition of CXC chemokine receptor 3" for publication as an Article in Nature Structural & Molecular Biology.

Your paper will be published online soon after we receive proof corrections and will appear in print in the next available issue. You can find out your date of online publication by contacting the production team shortly after sending your proof corrections. Content is published online weekly on Mondays and Thursdays, and the embargo is set at 16:00 London time (GMT)/11:00 am US Eastern time (EST) on the day of publication. Now is the time to inform your Public Relations or Press Office about your paper, as they might be interested in promoting its publication. This will allow them time to prepare an accurate and satisfactory press release. Include your manuscript tracking number (NSMB-A47307B) and our journal name, which they will need when they contact our press office.

About one week before your paper is published online, we shall be distributing a press release to news organizations worldwide, which may very well include details of your work. We are happy for your institution or funding agency to prepare its own press release, but it must mention the embargo date and Nature Structural & Molecular Biology. If you or your Press Office have any enquiries in the meantime, please contact press@nature.com.

Please note that *Nature Structural & Molecular Biology* is a Transformative Journal (TJ). Authors may publish their research with us through the traditional subscription access route or make their paper immediately open access through payment of an article-processing charge (APC). Authors will not be required to make a final decision about access to their article until it has been accepted. Find out more about Transformative Journals <https://www.springernature.com/gp/open-research/transformative-journals>

Authors may need to take specific actions to achieve [open access](https://www.springernature.com/gp/open-research/funding/policy-)

compliance-faqs"> compliance with funder and institutional open access mandates.

If your research is supported by a funder that requires immediate open access (e.g. according to [Plan S principles](https://www.springernature.com/gp/open-research/plan-s-compliance)) then you should select the gold OA route, and we will direct you to the compliant route where possible. For authors selecting the subscription publication route, the journal's standard licensing terms will need to be accepted, including [self-archiving policies](https://www.springernature.com/gp/open-research/policies/journal-policies). Those licensing terms will supersede any other terms that the author or any third party may assert apply to any version of the manuscript.

Sincerely,

Katarzyna Ciazynska
(she/her)
Associate Editor
Nature Structural & Molecular Biology
<https://orcid.org/0000-0002-9899-2428>

Click here if you would like to recommend Nature Structural & Molecular Biology to your librarian:

<http://www.nature.com/subscriptions/recommend.html#forms>